# DON'T FLATTEN, TOKENIZE! UNLOCKING THE KEY TO SOFTMOE'S EFFICACY IN DEEP RL

**Ghada Sokar**
Google DeepMind
gsokar@google.com

**Johan Obando-Ceron**
Mila, Université de Montréal
jobando0730@gmail.com

**Aaron Courville**
Mila, Université de Montréal
courvila@mila.quebec

**Hugo Larochelle**
Google DeepMind
hugolarochelle@google.com

**Pablo Samuel Castro**
Google DeepMind
Mila, Université de Montréal
psc@google.com

## ABSTRACT

The use of deep neural networks in reinforcement learning (RL) often suffers from performance degradation as model size increases. While soft mixtures of experts (SoftMoEs) have recently shown promise in mitigating this issue for online RL, the reasons behind their effectiveness remain largely unknown. In this work we provide an in-depth analysis identifying the key factors driving this performance gain. We discover the surprising result that tokenizing the encoder output, rather than the use of multiple experts, is what is behind the efficacy of SoftMoEs. Indeed, we demonstrate that even with an appropriately scaled single expert, we are able to maintain the performance gains, largely thanks to tokenization.

## 1 INTRODUCTION

Deep networks have been central to many of the successes of reinforcement learning (RL) since their initial use in DQN (Mnih et al., 2015a), which has served as the blueprint (in particular with regards to network architecture) for most of the followup works. However, recent works have highlighted the need to better understand the network architectures used, and question whether alternate

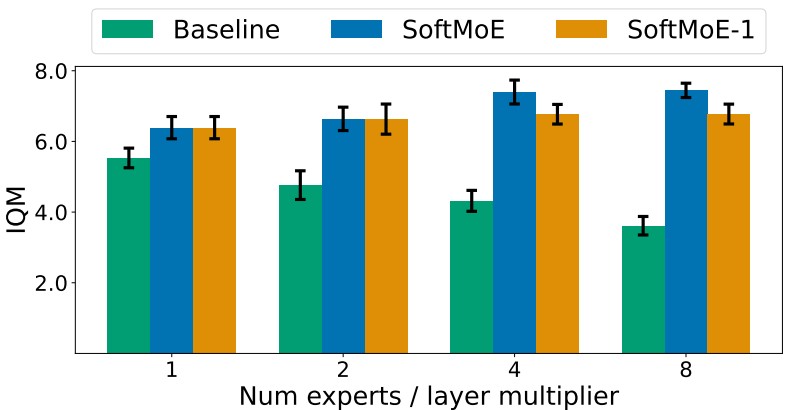

Figure 1: Comparison of IQM (Agarwal et al., 2021) for Rainbow (Hessel et al., 2018), using the Impala architecture (Espeholt et al., 2018), with penultimate layer scaled, and with SoftMoE with varying numbers of experts, and a single, scaled, expert. IQM scores computed over 200M environment steps across 20 games, with 5 independent runs each, higher is better. Error bars represent 95% stratified bootstrap confidence intervals. A single scaled expert matches the performance of multiple experts.

choices can be more effective (Graesser et al., 2022; Sokar et al., 2022; 2023; Obando Ceron et al., 2024; Lee et al., 2025). Indeed, Obando Ceron* et al. (2024) and Willi* et al. (2024) demonstrated that the use of Mixtures-of-Experts (MoEs) (Shazeer et al., 2017), and in particular soft mixtures-of-experts (SoftMoEs) (Puigcerver et al., 2024), yields models that are more parameter efficient. Obando Ceron* et al. (2024) hypothesized that the underlying cause of the effectiveness of SoftMoEs was in the induced *structured sparsity* in the network architecture. While some of their analyses did seem to support this, the fact that performance gains were observed *even with a single expert* seems to run counter to this hypothesis.

In this work we seek to better understand the true underlying causes for the effectiveness of Soft-MoEs in RL settings. Specifically, we perform a careful and thorough investigation into the many parts that make up SoftMoEs aiming to isolate the impact of each. Our investigation led to the surprising finding that **the core strength of SoftMoEs lies in the tokenization of the outputs of the convolutional encoder**.

The main implications of this finding are two-fold. On the one hand, it suggests that we are still under-utilizing experts when incorporating MoEs in deep RL, given that we are able to obtain comparable performance with a single expert (see Figure 1). On the other hand, this finding has broader implications for deep RL agents trained on pixel-based environments, suggesting that the common practice of flattening the multi-dimensional outputs of the convolutional encoders is ineffective.

Our main contributions can be summarized as follows:

- We conduct a series of analyses to identify the key factors that allow soft Mixture of Experts (SoftMoEs) to effectively scale RL agents.
- We demonstrate that tokenization plays a crucial role in the performance improvements.
- We demonstrate that experts within the SoftMoEs are not specialized to handle specific subsets of tokens and exhibit redundancy. We explore the use of recent techniques for improving network utilization and plasticity to mitigate this redundancy.

The paper is organized as follows. We introduce the necessary background for RL in Section 2 and MoEs in Section 3; we present our experimental setup in Section 4.1, and present our findings from Section 4.2 to Section 6; finally, we discuss related works in Section 7 and provide a concluding discussion in Section 8.

## 2   ONLINE DEEP REINFORCEMENT LEARNING

In online reinforcement learning, an agent interacts with an environment and adjusts its behaviour based on the reinforcement (in the form of rewards or costs) it receives from the environment. At timestep $t$, the agent inhabits a state $x_t \in \mathcal{X}$ (where $\mathcal{X}$ is the set of all possible states) and selects an action $a_t \in \mathcal{A}$ (where $\mathcal{A}$ is the set of all possible actions) according to its behaviour policy $\pi : \mathcal{X} \to \mathfrak{P}(\mathcal{A})$; the environment's dynamics yield a new state $x_{t+1} \in \mathcal{X}$ and a numerical reward $r_t \in \mathbb{R}$. The goal of an RL agent is to find a policy $\pi^*$ that maximizes $V^\pi(x) := \sum_{t=0}^\infty [\gamma^t r_t | x_0 = x, a_t \sim \pi(x_t)]$, where $\gamma \in [0, 1)$ is a discount factor. In value-based RL, an agent maintains an estimate of $Q^\pi : \mathcal{X} \times \mathcal{A} \to \mathbb{R}$, defined as $Q^\pi(x, a) = \sum_{t=0}^\infty [\gamma^t r_t | x_0 = x, a_0 = a, a_t \sim \pi(x_t)]$; from any state $x \in \mathcal{X}$, a policy $\pi$ can then be improved via $\pi(x) = \arg\max_{a \in \mathcal{A}} Q^\pi(x, a)$ (Sutton & Barto, 1998).

In its most basic form, value-based deep reinforcement learning (DRL) extends this approach by using neural networks, parameterized by $\theta$, to estimate $Q$. While the network architectures used may vary across environments, the one originally introduced by Mnih et al. (2015a) for DQN has served as the basis for most. Given that its inputs consist of frames of pixels from Atari games (Bellemare et al., 2013), this network consisted of a series of convolutional layers followed by a series of fully-connected layers; for the remainder of this work we will refer to the convolutional layers as the *encoder*. The top panel of Figure 2 displays a simplified version of this network, where the encoder produces a 3-dimensional tensor with dimensions $(h, w, d)$; this tensor is flattened to a vector of length $h * w * d$, which is fed through fully-connected layers to produce the estimates for $Q$. One notable variant to the original DQN encoder is Impala's use of a ResNet encoder (Espeholt et al., 2018), which has been shown to outperform the former. In this work, we mainly focus on the more recent ResNet encoder, but also include some results using the original DQN encoder.

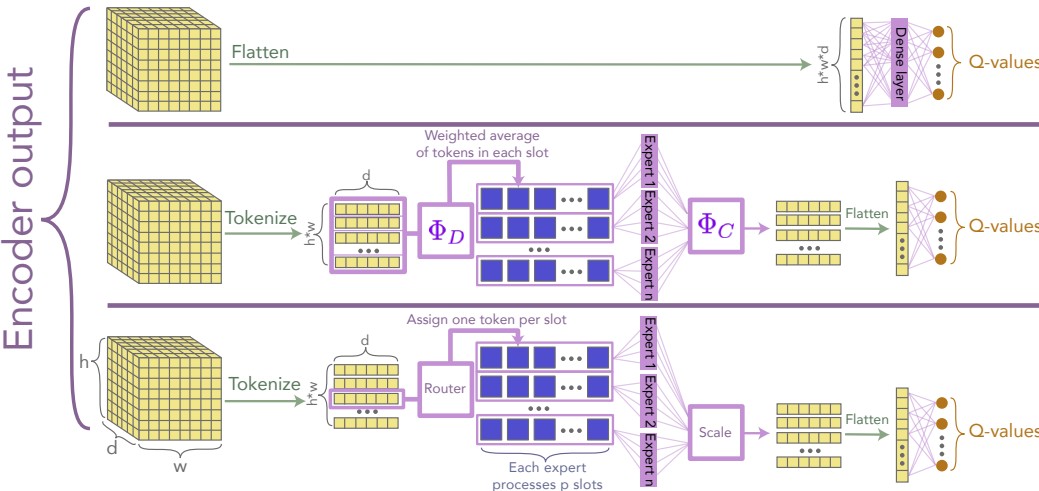

Figure 2: **The various architectures considered in this work.** *Top:* baseline architecture; *Middle:* SoftMoE architecture; *Bottom:* Top-$k$ architecture.

## 3 MIXTURES OF EXPERTS

While there have been a number of architectural variations proposed, in this work we focus on Mixtures of Experts as used by Obando Ceron* et al. (2024). Mixtures of experts (**MoEs**) were originally proposed as a network architecture for supervised learning tasks using conditional computation, which enabled training very large networks (Shazeer et al., 2017; Fedus et al., 2022). This was achieved by including MoE modules which route a set of input *tokens* to a set of $n$ sub-networks, or *experts*; the output of these experts are then combined and fed through the rest of the outer network. The routing/gating mechanism and the expert sub-networks are all trainable. In top-$k$ routing, we consider two methods for pairing tokens and experts. In **token choice routing**, each token gets an assignation to $k$ experts, each of which has $p$ *slots* (the number of processed inputs by each expert); this may result in certain tokens being dropped entirely by the MoE module and, more critically, certain experts may receive no tokens (Gale et al., 2023). To overcome having load-imbalanced experts, Zhou et al. (2022) proposed **expert choice routing**, where each expert chooses $p$ tokens for its $p$ slots, which results in better expert load-balancing. See the bottom row of Figure 2 for an illustration of this. Unless otherwise specified, we use a value of $p$ equals to $\frac{\#tokens}{\#experts}$. In most of our analyses, we focus on expert choice routing as it demonstrates higher performance than token choice routing (see Appendix B.1 for full empirical comparison between the two across varying number of experts).

Puigcerver et al. (2024) introduced **SoftMoEs**, which replace the hard token-expert pairing with a "soft" assignation. Specifically, rather than having each expert slot assigned to a specific token, each slot is assigned a (learned) weighted combination of all the tokens. Thus, each expert receives $p$ weighted averages of all the input tokens. See the middle row of Figure 2 for an illustration of this.

Obando Ceron* et al. (2024) proposed replacing the penultimate dense layer of a standard RL network with an MoE module; the authors demonstrated that, when using SoftMoEs, value-based online RL agents exhibit signs of a "scaling law", whereby adding more parameters (in the form of more experts) monotonically improves performance. This is in stark contrast with what is observed with the original architectures, where extra parameters result in performance degradation (see Figure 1). The authors argued that the structured sparsity induced by the MoE architecture is what was primarily responsible for their strong performance.

In order to maintain the shape of the input tokens (as originally done with MoE architectures), Obando Ceron* et al. (2024) add a projection layer after the "combine" step ($\Phi_C$ in Figure 2). Another important detail is the proposed tokenization scheme. MoEs are most commonly used in NLP transformer architectures where tokens are clearly defined (typically as encodings of words),

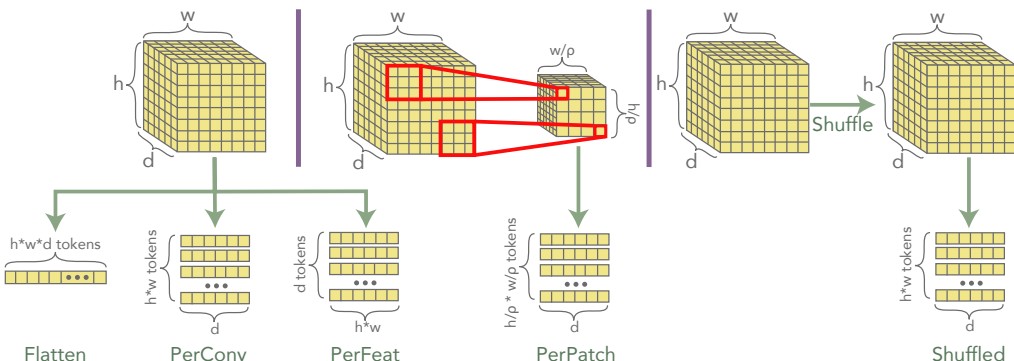

Figure 3: **Tokenization schemes considered in this work**, specified in the bottom row.

but this tokenization is not something readily available for online RL agents. Thus, Obando Ceron* et al. (2024) proposed a number of mechanisms for tokenizing the output of the encoder.

**Tokenization**  As mentioned in section 2, the output of a standard RL encoder is a 3-dimensional tensor with dimensions $(h, w, d)$; Obando Ceron* et al. (2024) proposed a few possible ways of creating tokens from this output. **PerConv** tokenization creates $h*w$ tokens of dimension $d$; **PerFeat** tokenization creates $d$ tokens of dimension $h*w$; see Figure 3 for an illustration. Although necessary for compatibility with transformer MoEs, this tokenization is a departure from the standard practice of flattening the encoder output, as discussed in Section 2.

## 4    UNDERSTANDING THE IMPACT OF THE SOFTMOE COMPONENTS

While Obando Ceron* et al. (2024) demonstrated the appeal of SoftMoEs for online RL agents, it was not clear whether all of the components of SoftMoEs were important for the performance gains, or whether some were more critical. Further, it was observed that SoftMoEs failed to provide similar gains in actor-critic algorithms such as PPO (Schulman et al., 2017) and SAC (Haarnoja et al., 2018). This section aims to isolate the impact of each of these components.

### 4.1    EXPERIMENTAL SETUP

The genesis of our work is to provide a deeper understanding of SoftMoEs in online RL. For this reason, we will mostly focus on SoftMoEs applied to the Rainbow agent (Hessel et al., 2018) with 4 experts and the Impala achitecture (Espeholt et al., 2018), as this was the most performant setting observed by Obando Ceron* et al. (2024). We explore variations of this setting where useful in clarifying the point being argued. We evaluate performance on the same 20 games of the Arcade Learning Environment (ALE) (Bellemare et al., 2013) benchmark used by Obando Ceron* et al. (2024) for direct comparison, with 5 independent seeds for each experiment. However, we include the results of our main findings on the full suite of 60 games. We will use the notation "SoftMoE-$n$" to denote a SoftMoE architecture with $n$ experts. All experiments were run on Tesla P100 GPUs using the same Dopamine library (Castro et al., 2018) used by Obando Ceron* et al. (2024); each run with 200 million environment steps took between six and eight days. Following the guidelines suggested by Agarwal et al. (2021), we report human-normalized aggregated interquartile mean (IQM), regular mean, median, and optimality gap, with error bars indicating 95% stratified bootstrap confidence intervals. Performance gains are indicated by increased IQM, median, and mean scores, and reduced optimality gaps. See Appendix A for further details.

### 4.2    ANALYSIS OF SOFTMOE COMPONENTS

At a high level, SoftMoE modules for online RL agents consist of **(i)** the use of a learnable tensor $\Phi$ used to obtain the dispatch ($\Phi_D$) and combine ($\Phi_C$) weights; **(ii)** processing $p$ input slots per expert; **(iii)** the architectural dimensions (network depth and width); **(iv)** the use of $n$ experts; and **(v)** the tokenization of the encoder output. The middle row of Figure 2 provides an illustration of these. In

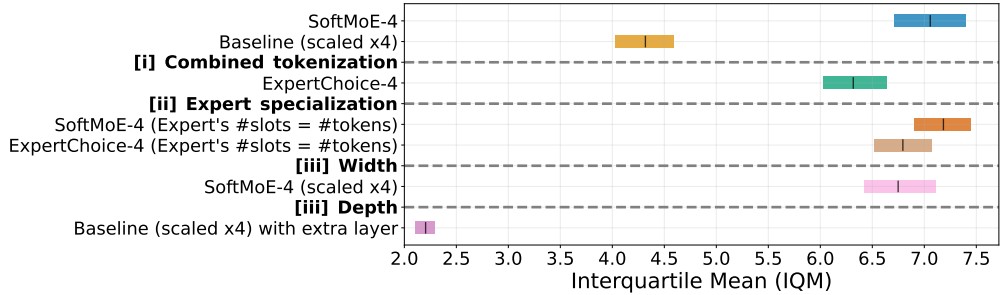

Figure 4: **Understanding the impact of SoftMoE components.** Using Rainbow with the Impala architecture as our base, we investigate key aspects of SoftMoE-4: (i) combining tokens, (ii) expert specialization, and (iii) adjusting architectural dimensions. Reporting IQM (Agarwal et al., 2021), where higher is better. SoftMoE does not appear to suffer from the performance degradation observed in the baseline, even when increasing its width.

this section we discuss the first four with empirical evaluation demonstrated in Figures 1 and 4, and devote a separate section for tokenization, which turns out to be crucial for performance. The same analysis was performed on DER, yielding the same results, and can be found in Appendix B.3.

**Processing *combined* tokens** To assess the impact of SoftMoE's weighted combination of tokens via the learnable weights $\Phi$, we compare it with expert choice routing, where experts select sets of (non-combined) tokens. While combined tokens yield improved performance, expert choice routing itself demonstrates substantial gains over the baseline and effectively scales the network (see group **[i]** in Figure 4). Thus, *combined tokens alone do not explain SoftMoE's efficacy.*

**Expert specialization** Each expert in a MoE architecture has a limited set of *slots*: with token/expert selection each slot corresponds to one token, while in SoftMoE each slot corresponds to a particular weighted combination of all tokens. This constraint could encourage each expert to specialize in certain types of tokens in a way that is beneficial for performance and/or scaling. To investigate, we increase the number of slots to be equal to the number of tokens (effectively granting unlimited capacity). Group **[ii]** in Figure 4 demonstrates that for both SoftMoE and expert choice routing, the performance sees little change when increasing the number of slots, suggesting that *expert specialization is not a major factor in SoftMoE's performance.*

**Expert width** Obando Ceron* et al. (2024) scaled up the penultimate layer of the baseline model with the intent of matching the parameters of the SoftMoE networks, thereby demonstrating that naïvely scaling width degrades performance. While they demonstrated that a performance drop is not observed when scaling *down* the dimensionality of each expert, they did not investigate what occurs when scaling *up* the experts dimensionality. This experiment makes sense if comparing the baseline's penultimate layer dimensionality with that of each expert, rather than with the *combined* experts dimensionality. Scaling up the width of experts in SoftMoE does not lead to the same performance degradation observed in the scaled baseline model as shown in the first group **[iii]** of Figure 4.

**Network depth** As mentioned in Section 3, SoftMoE models have an additional projection layer within each expert to maintain input token size, resulting in a different number of layers compared to the baseline model. To determine whether this extra layer contributes to SoftMoE's performance gains, we add a layer of the same size to the baseline model after the penultimate layer. The second group **[iii]** in Figure 4 demonstrates that this added layer does not improve the baseline model's performance nor its ability to scale, suggesting that *SoftMoE's performance gains are not attributable to the presence of the extra projection layer.*

**Multiple experts** Obando Ceron* et al. (2024) demonstrated that multiple experts improves performance. However, the performance gains plateau beyond a certain point, with minimal improvement observed after utilizing four experts (see Figure 1). Consequently, the presence of multiple experts cannot fully account for the observed performance advantages over the baseline model, suggesting that other factors are also contributing to the improved results.

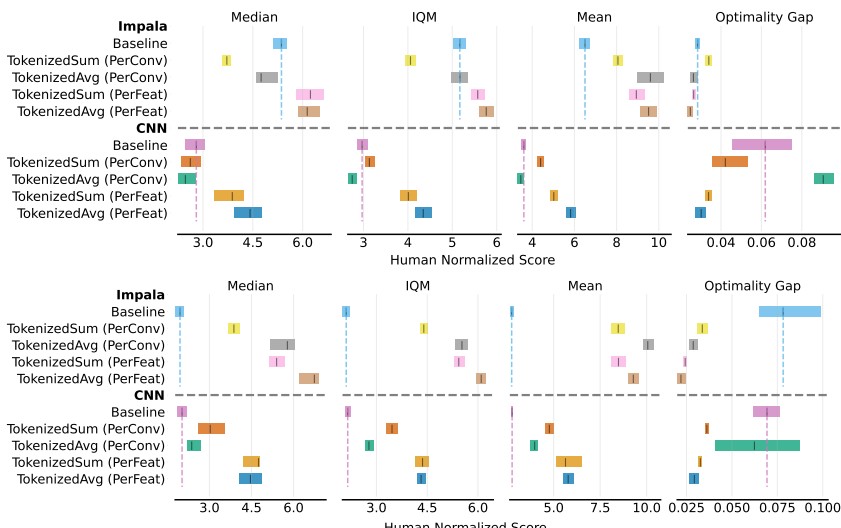

Figure 5: **Evaluating impact of tokenizing Rainbow-lite** with unscaled (top) and scaled (bottom) architectures, using sum/mean over convolutional features, and exploring both PerConv and PerFeat tokenization schemes. Reporting Median, IQM, Mean, and Optimaly Gap scores (Agarwal et al., 2021). Higher is better for all except Optimality Gap. Tokenization with a scaled representation can yield significant significant performance gains. Per-game results available in Appendix C.1.

## 5 DON'T FLATTEN, TOKENIZE!

A crucial structural difference remains in how SoftMoE and the baseline model process the encoder's output: while the baseline model flattens the output of the final convolutional layer, SoftMoE divides it into distinct tokens (see Section 3). We conduct a series of experiments to properly assess the impact of this choice, going from the standard baseline architecture to a single-expert SoftMoE.

**Tokenized baseline**    Our first experiment simply adds tokenization to the baseline architecture. Specifically, given the 3-dimensional encoder output with shape $[h, w, d]$, we replace the standard flattening (to a vector of dimension $h*w*d$) with a reshape to a matrix of shape $[h*w, d]$ for PerConv tokenization or $[d, h*w]$ for PerFeat; then, prior to the final linear layer, we either sum over the first dimension or take their average. Since this operation is not immediately applicable to Rainbow (due to Noisy networks), we explored this experiment with Rainbow-lite, a version of Rainbow in Dopamine which only includes C51 (Bellemare et al., 2017), multi-step returns, and prioritized experience replay. Figure 5 demonstrates that this simple change can yield statistically significant improvements for Rainbow-lite, *providing strong evidence that tokenization plays a major role in the successful scaling of DRL networks*.

**Single expert processing all tokens sparsely**    To assess the influence of tokenization within the MoE framework, while ablating all other four components studied in Section 4.2, we explore using a *single* expert to process *all* input tokens. Each token is *sparsely* processed in one of the single expert slots. This corresponds to expert choice-1. In line with the "expert width" investigation from the last section, we study expert choice-1 with the scaled expert dimensionality to match the scaled baseline one. As Figure 6 shows, even with a single expert the performance is comparable to that of SoftMoE with multiple experts. This suggests *that tokenization is one of the major factors in the performance gains achieved by SoftMoE.*

**Combined tokenization is a key driver**    We further investigate the impact of *combined* tokenization in the single expert setting. To this end, we study SoftMoE-1, with scaled expert dimensionality as before. As Figure 6 shows, using combined tokens with a single expert brings the performance even closer to the multiple expert setting. Similar results were observed in SoftMoE with 2 and 8

experts (See Appendix B.2 for more details). This further confirms that *(combined) tokenization is the key driver of SoftMoE's efficacy*.

## 5.1 ADDITIONAL ANALYSES

The main takeaway from the previous section is that SoftMoE with a single (scaled) expert is as performant as the variant with multiple unscaled experts. To better understand this finding, in this section we perform an extra set of experiments with this variant of SoftMoE.

**Are all types of tokenization equally effective?** We compare the different tokenization types presented in section 3, in addition to two new types of tokenization, which serve as useful references: **PerPatch** tokenization splits the 3-dimensional $[h, w, d]$ tensor into $\rho$ patches, over which average pooling is performed resulting in a new tensor of dimensions $[h/\rho, w/\rho, d]$; **Shuffled** is similar to PerConv, except it first applies a fixed permutation on the encoder output. See Figure 3 for an illustration of these. The comparison of the different tokenization types is summarized in Figure 7. Consistent with the findings of Obando Ceron* et al. (2024), PerConv performs best, and in particular outperforms PerFeat. It is interesting to observe that PerPatch is close in performance to PerConv, while Shuffled is the worst performing. The degradation from PerPatch is likely due to the average pooling and reduced dimensionality; the degradation from shuffling is clearly indicating that it is important to maintain the spatial structure of the encoding output. Combining these results with our previous findings suggests that *using tokenization maintains the spatial structure learned by the convolutional encoder, and is crucial to maximizing performance*.

**Computational efficiency by combined tokens** Puigcerver et al. (2024) highlighted that the time complexity of SoftMoE depends directly on the number of slots. As can be seen in Figure 8, using SoftMoE-1 with only 10% of the default number of slots (kept fixed throughout training), and even with a single slot, achieves a comparable performance to using the regular number of slots. On the other hand, the reduced performance of ExpertChoice-1 further confirms the gains added by combining tokens in SoftMoE.

**More games and different encoders** Our results so far were run on the 20 games used by Obando Ceron* et al. (2024) and with the Impala ResNet encoder (Espeholt et al., 2018). To ensure our findings are not specific to these choices, we extend our evaluation to all 60 games in the ALE suite, and with the standard CNN encoder of Mnih et al. (2015b). Our findings, as illustrated in Figure 9, demonstrate consistent results across all games and various encoder architectures. We further conducted extra experiments on Procgen (Cobbe et al., 2019), which are consistent with our findings (Figure 21 in Appendix C).

**More agents** We focused on Rainbow for the majority of our investigation due to the positive results demonstrated by Obando Ceron* et al. (2024), but it is important to also consider other agents. To this point, we evaluate the impact of a single scaled expert on DQN (Mnih et al., 2015b)

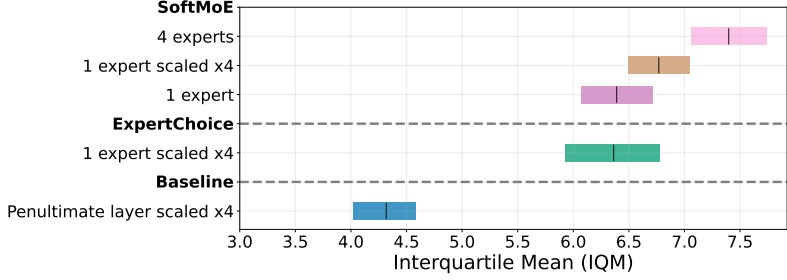

Figure 6: **Combined tokenization is a key driver of SoftMoE's efficacy**. Reporting IQM (Agarwal et al., 2021), where higher is better. SoftMoE with a single scaled expert has comparable performance with using multiple experts. Even *non-combined* tokenization (e.g. ExpertChoice) with a single expert achieves similar performance.

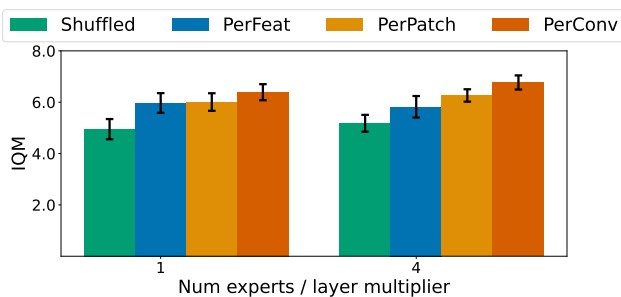

Figure 7: **Comparison of different tokenization schemes** for SoftMoE-1 with the standard and scaled penultimate layer. Reporting IQM (Agarwal et al., 2021), where higher is better. PerConv achieves the highest performance, suggesting that maintaining the spatial structure of the features plays a role on the observed benefits.

and DER (Van Hasselt et al., 2019). Note that we replace the original Huber loss used in DQN with mean-squared error loss, as it has been shown to be more performant (Ceron & Castro, 2021), and coincides with what was used by Obando Ceron* et al. (2024). Following the training regime used by Obando Ceron* et al. (2024), we train DER for 50M environment frames. In the DQN setting, SoftMoE-1 does not appear to offer significant improvements over the baseline, potentially due to the limited gains observed with SoftMoE-4 (Figure 10, top). Further investigation is needed to understand how the inherent instability of DQN might limit the potential benefits of SoftMoE. On the other hand, consistent with our finding in Rainbow, in DER, where SoftMoE-4 achieves a superior performance over the baseline, SoftMoE-1 is able to offer the same gains (Figure 10, bottom). To further evaluate the generality of our claims, we conducted initial experiments with SAC (Haarnoja et al., 2018) on the CALE (Farebrother & Castro, 2024), which is a continuous action version of the original ALE. In Figure 22 in Appendix C it can be seen that tokenization provides an improvement over flattening.

## 6   CALLING ALL EXPERTS!

Our findings in Sections 4 and 5 reveal that the main benefit of using SoftMoE architectures is in the tokenization and their weighted combination, rather than in the use of multiple experts. The fact that we see no evidence of expert specialization suggests that most experts are redundant (see Figure 4). Rather than viewing this as a negative result, it's an opportunity to develop new techniques to improve expert utilization and maximize the use of MoE architectures in RL.

As a first step in this direction, we adopt existing techniques designed to improve network utilization: (1) periodic *reset* of parameters (Nikishin et al., 2022) and (2) Shrink and Perturb (S&P) (Ash & Adams, 2020), as used by D'Oro et al. (2022) and Schwarzer* et al. (2023). We perform these experiments on Rainbow and DER with SoftMoE-4. We explore applying resets and S&P to either all experts or a subset of experts, where the experts to reset are those that have the highest fraction of

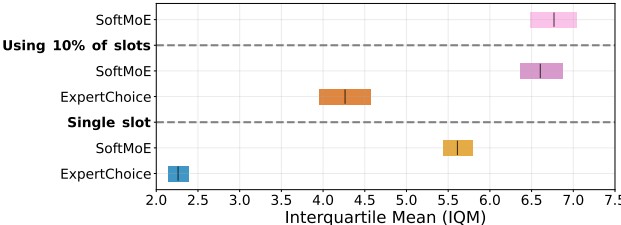

Figure 8: **Evaluating the impact of fewer slots with a single scaled (by 4x) expert** for both SoftMoE and ExpertChoice. Reporting IQM (Agarwal et al., 2021), where higher is better. Even with drastically fewer slots, SoftMoE-1 remains competitive.

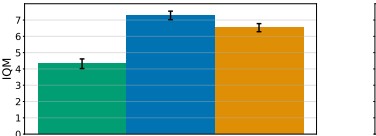 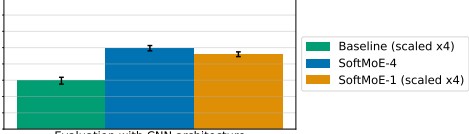

Figure 9: (Left) **Evaluating the performance on all of the 60 games of Atari 2600**. Reporting IQM (Agarwal et al., 2021), where higher is better. Consistent with our results SoftMoE-1 matches the performance of multiple experts over the full suite. (Right) **CNN Encoder**. The performance gain of SoftMoE-1 is present when training with the standard CNN encoder, revealing the importance of tokenization across architectures.

dormant neurons (Sokar et al., 2023). We reset every 20M and 200K environment steps for Rainbow and DER, respectively. More experimental details can be found in Appendix B.5.

Interestingly, the effectiveness of these techniques varied across different RL agents. While weight reset and S&P did not improve expert utilization in the Rainbow agent, they led to enhanced performance in the sample-efficient DER agent (Figure 11). Within the DER agent, resetting a subset of experts proved more beneficial than resetting all, while applying S&P to all experts yielded the most significant performance gains. These findings highlight the need for further research into methods that can optimize expert utilization and improve performance in RL with SoftMoE.

## 7 RELATED WORKS

**Mixture of Experts**   Mixture of Experts (MoEs) have demonstrated remarkable success in scaling language and computer vision models to trillions of parameters (Lepikhin et al., 2020; Fedus et al., 2022; Wang et al., 2020; Yang et al., 2019; Abbas & Andreopoulos, 2020; Pavlitskaya et al., 2020). Chung et al. (2024) examined the implicit biases arising from token combinations in SoftMoE and defined the notion of expert specialization in computer vision. Besides the success of MoEs in single task training, similar gains have been observed in transfer and multitask learning (Puigcerver et al., 2020; Chen et al., 2023; Ye & Xu, 2023; Hazimeh et al., 2021). In addition to the works of Obando Ceron* et al. (2024) and Willi* et al. (2024) already discussed, Farebrother et al. (2024) proposed replacing regression loss with classification loss and showed its benefits on MoEs.

**Network scalability in RL**   Scaling reinforcement learning (RL) networks presents significant challenges, largely due to issues with training instabilities (Hessel et al., 2018). To address these difficulties, several techniques have been proposed, with a focus on algorithmic improvements or hyperparameter optimization (Farebrother et al., 2022; 2024; Taiga et al., 2022; Schwarzer* et al., 2023). Recently, a few studies have begun to focus on architectural innovations as a means of addressing these challenges. Kumar et al. (2023a) showed that incorporating a learned spatial embedding with the output of a ResNet encoder, prior to flattening it, improves the scalability in offline RL. Obando Ceron et al. (2024) demonstrated that gradual magnitude pruning enables the scaling of value-based agents, resulting in substantial performance gains.

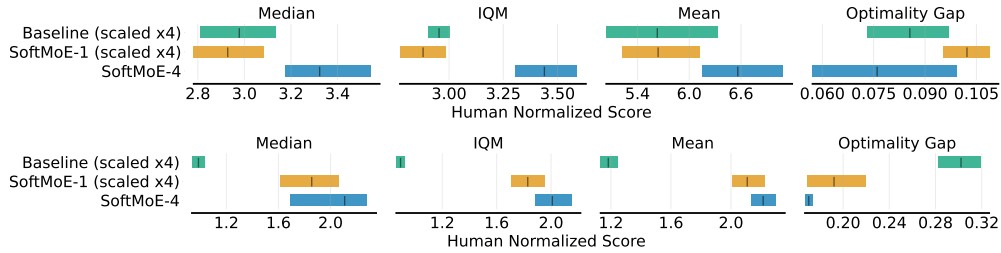

Figure 10: **Evaluating on DQN (top) and DER (bottom).** Reporting Median, IQM, Mean, and Optimaly Gap scores (Agarwal et al., 2021). Higher is better for all except Optimality Gap. When combined with DQN, SoftMoE with a single scaled expert does not provide gains over the baseline. In contrast, when combined with DER, it achieves performance comparable to SoftMoE-4.

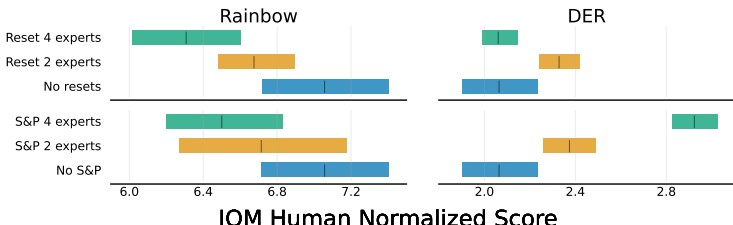

Figure 11: **Evaluating the use of resets (top) and S&P (bottom) for improving expert utilization on Rainbow (left column) and DER (right column).** Reporting IQM (Agarwal et al., 2021), where higher is better. All experiments run with SoftMoE-4. Resets and S&P seem to benefit DER, but degrade performance for Rainbow.

**Network plasticity in RL**   It has been observed that networks trained on non-stationary distributions tend to lose their expressivity over time, a phenomenon referred to as the loss of plasticity (Kumar et al., 2021; Lyle et al., 2021). While several studies have attempted to explain this phenomenon (Lyle et al., 2023; Ma et al., 2023; Lyle et al., 2024), the underlying causes remain uncertain. In a parallel line of research, numerous methods have been proposed to improve network trainability. These approaches involve using small batch size for gradient updates (Ceron et al., 2023), periodically resetting some of the network's parameters to random values (Sokar et al., 2023; Nikishin et al., 2022; Igl et al., 2020; Dohare et al., 2024), slightly modifying the existing weights by shrinking and perturbing them (D'Oro et al., 2022; Schwarzer* et al., 2023), and restricting the large deviation from initial weights (Kumar et al., 2023b; Lewandowski et al., 2024). Recently, Lee et al. (2024) have also demonstrated that combining reinitialization with a teacher-student framework can enhance plasticity. Another technique involves preserving useful neurons by perturbing gradients to ensure smaller updates for important neurons (Elsayed & Mahmood, 2024). Liu et al. (2025) proposed dynamic network growth through neuron generation and dormant neuron pruning. In this work, we explore some of these methods to improve plasticity and utilization of experts.

## 8   CONCLUSIONS

We began our investigation by evaluating the components that make up a SoftMoE architecture, so as to better understand the strong performance gains reported by Obando Ceron* et al. (2024) when applied to value-based deep RL agents. One of the principal, and rather striking, findings is that tokenizing the encoder outputs is one of the primary drivers of the performance improvements. Since tokenization is mostly necessary for pixel-based environments (where encoders typically consist of convolutional layers), it can explain the absence of performance gains observed by Obando Ceron* et al. (2024) and Willi* et al. (2024) when using MoE architectures in actor-critic algorithms, since these were evaluated in non-pixel-based environments. Initial experiments illustrated in Figure 22 suggest that our findings carry over to actor-critic agents on continuous control environments.

More generally, our findings suggests that the common practice of flattening the outputs of deep RL encoders is sub-optimal, as it is likely *not* maintaining the spatial structure output by the convolutional encoders. Indeed, the results with PerConv versus Shuffled tokenization in Figure 7 demonstrate that maintaining this spatial structure is important for performance. This is further confirmed by our results in Figure 5, where we can observe that not flattening the convolutional outputs can lead to significant improvements. This idea merits further investigation, as the results vary depending on the architecture used and whether features are summed or averaged over.

It is somewhat surprising that our results do not seem to carry over to DQN. However, the fact that we *do* see similar results with Rainbow-lite and DER (which is based on Rainbow) suggests that the use of categorical losses may have a complementary role to play. This is a point argued by Farebrother et al. (2024) and merits further inquiry.

Although the use of mixtures-of-experts architectures for RL show great promise, our results suggest that most of the experts are not being used to their full potential. The findings in Figure 11 present one viable mechanism for increasing their use, but further research is needed to continue increasing their efficacy for maximally performant RL agents.

ACKNOWLEDGEMENTS

The authors would like to thank Gheorghe Comanici, Jesse Farebrother, Joan Puigcerver, Doina Precup, and the rest of the Google DeepMind team for valuable feedback on this work. We would also like to thank the Python community (Van Rossum & Drake Jr, 1995; Oliphant, 2007) for developing tools that enabled this work, including NumPy (Harris et al., 2020), Matplotlib (Hunter, 2007) and JAX (Bradbury et al., 2018).

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

## A   EXPERIMENTAL DETAILS

**Tasks**   We evaluate Rainbow on 20 games from the Arcade Learning Environment (Bellemare et al., 2013). We use the same set of games evaluated by Obando Ceron* et al. (2024) for direct comparison. This includes: Asterix, SpaceInvaders, Breakout, Pong, Qbert, DemonAttack, Seaquest, WizardOfWor, RoadRunner, BeamRider, Frostbite, CrazyClimber, Assault, Krull, Boxing, Jamesbond, Kangaroo, UpNDown, Gopher, and Hero. In addition, we run some of our main results on all suite of 60 games.

On Atari100k benchmark, we evaluate on the full 26 games. These are: Alien, Amidar, Assault, Asterix, BankHeist, BattleZone, Boxing, Breakout, ChopperCommand, CrazyClimber, DemonAttack, Freeway, Frostbite, Gopher, Hero, Jamesbond, Kangaroo, Krull, KungFuMaster, MsPacman, Pong, PrivateEye, Qbert, RoadRunner, Seaquest, UpNDown.

**Hyper-parameters**   We use the default hyper-parameter values for DQN (Mnih et al., 2015a), Rainbow (Hessel et al., 2018), and DER (Van Hasselt et al., 2019) agents. We share the details of these values in Table 1.

Table 1: Default hyper-parameters setting for DQN, Rainbow and DER agents.

|  | Atari | | |
| --- | --- | --- | --- |
| Hyper-parameter | DQN | Rainbow | DER |
| Adam's ($\epsilon$) | 1.5e-4 | 1.5e-4 | 0.00015 |
| Adam's learning rate | 6.25e-5 | 6.25e-5 | 0.0001 |
| Batch Size | 32 | 32 | 32 |
| Conv. Activation Function | ReLU | ReLU | ReLU |
| Convolutional Width | 1 | 1 | |
| Dense Activation Function | ReLU | ReLU | ReLU |
| Dense Width | 512 | 512 | 512 |
| Normalization | None | None | None |
| Discount Factor | 0.99 | 0.99 | 0.99 |
| Exploration $\epsilon$ | 0.01 | 0.01 | 0.01 |
| Exploration $\epsilon$ decay | 250000 | 250000 | 2000 |
| Minimum Replay History | 20000 | 20000 | 1600 |
| Number of Atoms | 0 | 51 | 51 |
| Number of Convolutional Layers | 3 | 3 | 3 |
| Number of Dense Layers | 2 | 2 | 2 |
| Replay Capacity | 1000000 | 1000000 | 1000000 |
| Reward Clipping | True | True | True |
| Update Horizon | 1 | 3 | 10 |
| Update Period | 4 | 4 | 1 |
| Weight Decay | 0 | 0 | 0 |
| Sticky Actions | True | True | False |

## B   ADDITIONAL EXPERIMENTS

### B.1   COMPARISON BETWEEN TOKEN CHOICE ROUTING AND EXPERT CHOICE ROUTING

We focus in our analysis on expert choice routing, where experts select tokens, as opposed to token choice routing, where tokens select experts. We compare these two routing strategies across different numbers of experts using a Rainbow agent with a ResNet architecture.

Figure 12 demonstrates that expert choice routing consistently outperforms token choice routing across varying number of experts. Moreover, this performance gap widens as the number of experts increases, highlighting the advantages of expert choice routing in addressing the limitations of token choice routing.

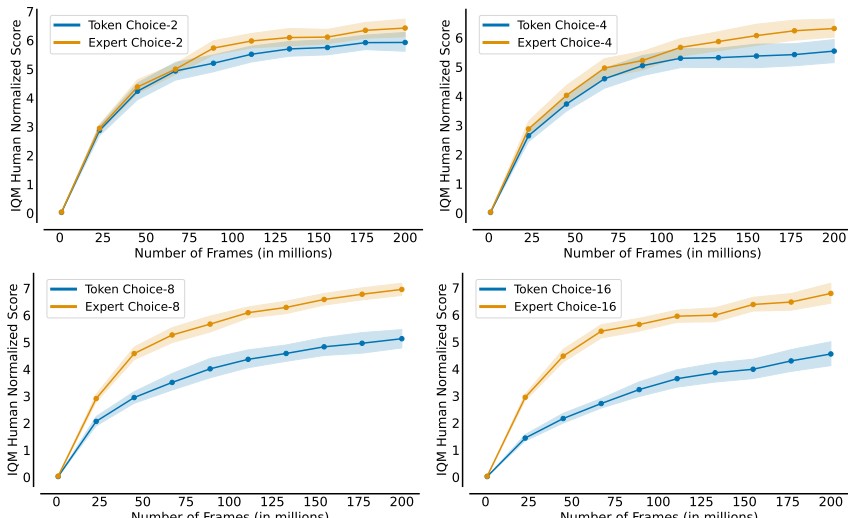

Figure 12: A comparison between token choice routing and expert choice routing on Rainbow with ResNet encoder. Expert choice routing consistently outperforms token choice routing across different number of experts. We report interquantile mean performance with error bars indicating 95% confidence intervals, over 20 games with 5 seeds each.

## B.2 DON'T FLATTEN TOKENIZE

In this section, we provide further analysis on the impact of tokenization and their weight combination on models with varying number of experts. To this end, we investigate SoftMoE-2 and SoftMoE-8 and compare it against the scaled SoftMoE-1 and ExpertChoice-1 (non-combined tokenization). Figure 13 shows that single scaled expert can reach a performance close to multiple experts, even in the case of large number of experts. In addition, consistent with our results, combined tokenization adds benefits over the sparse one. Figure 14 shows a comparison between the learning curves of single scaled expert and multiple ones.

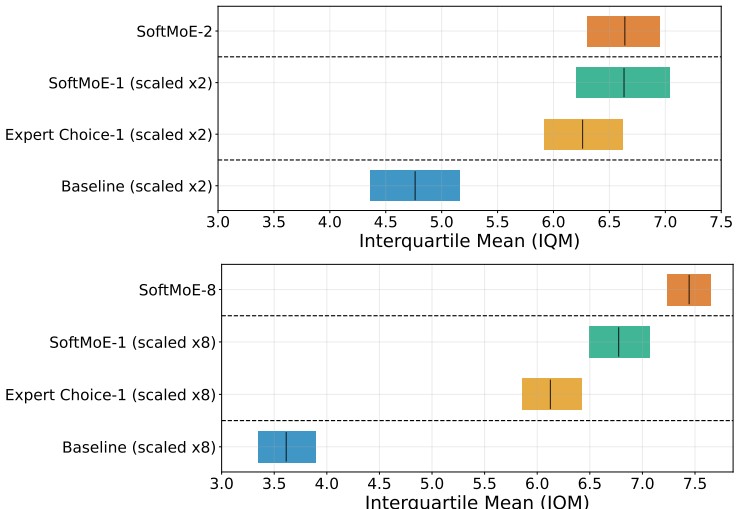

Figure 13: Combined tokenization plays a crucial role in performance. SoftMoE-1 with scaled hidden layer achieves a comparable performance to the standard SoftMoE.

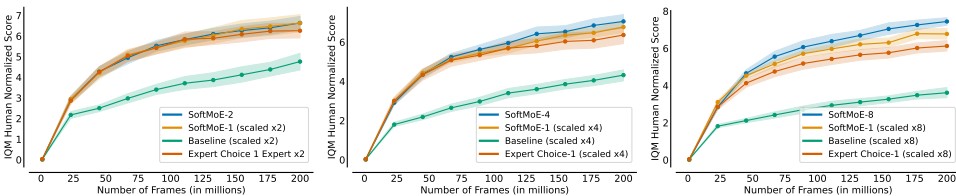

Figure 14: Comparison between the learning curves of the scaled SoftMoE-1 and multiple experts. We report interquantile mean performance with error bars indicating 95% confidence intervals, over 20 games with 5 seeds each.

To further assess the effect of flattening in reducing performance of agents, we replace the flattening operation in the baseline by global average pooling. We performed these experiments on the baseline with the default dimensionality and the scaled one in Rainbow.

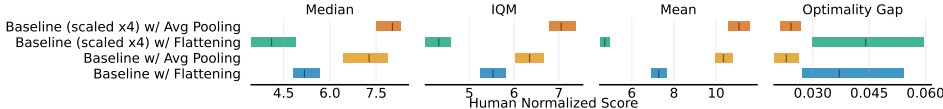

Figure 15: Replacing the flattening operation by global average pooling improves the performance of the baseline and enables scaling.

## B.3 ANALYSIS OF SOFTMOE COMPONENTS

In this section, we present an analysis of each of SoftMoE components on the DER algorithm. As shown in Figure 16, we observe similar tends to our findings in Section 4.2 on Rainbow. Combined tokens offers an additional benefits over the sparse ones. Experts do seem not be specialized in subset of the tokens. Adjusting the architectural depth and width for direct comparison with the baseline doesn't explain the performance gains of SoftMoE.

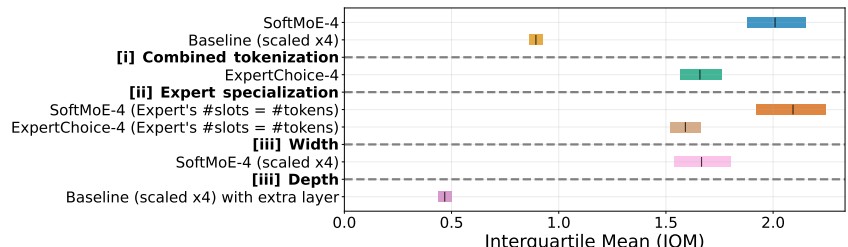

Figure 16: Understanding the impact of SoftMoE components on DER.

**Effect of Expert width**  In section 5, we studied the effect of scaling up the size of the expert hidden layer of SoftMoE-1. Here, we present the results of scaling down the size of this layer by 4. As shown in Figure 17, SoftMoE-1 with scaled down layer still outperform the baseline despite that it has less parameter count.

## B.4 MORE AGENTS

Same as our finding for DQN using SoftMoE-4, we observe that single expert has a closer performance to the scaled baseline than SoftMoE-8, as shown in Figure 18. Further investigation is needed to fully understand the behavior of SoftMoE on DQN.

## B.5 EXPERT UTILIZATION

In this section we provide more analysis to understand the utilization of experts in multi expert setting and the effect of existing techniques on improving plasticity.

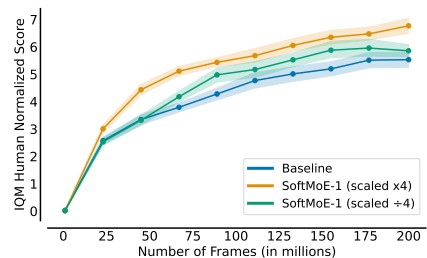

Figure 17: The effect of scaling down the size of the expert hidden layer in SoftMoE-1.

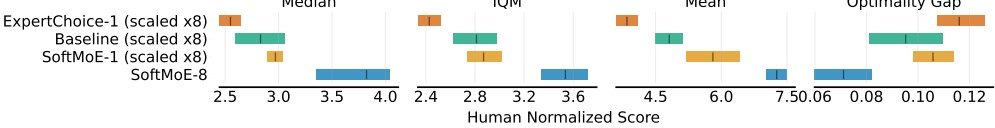

Figure 18: Evaluating more agents. When combined with DQN, SoftMoE with a single scaled expert does not provide large gains over the baseline.

### B.5.1 SOME EXPERTS ARE REDUNDANT

In Section 4.2, we showed that experts are not specialized in learning subset of tokens. We expand this analysis by assessing the redundancy of experts. To this end, we prune two experts of SoftMoE-4 during training. We studied two pruning schemes: pruning the two experts once and gradually prune throughout training. As shown in Figure 19, pruning these experts does not change the performance of the agent.

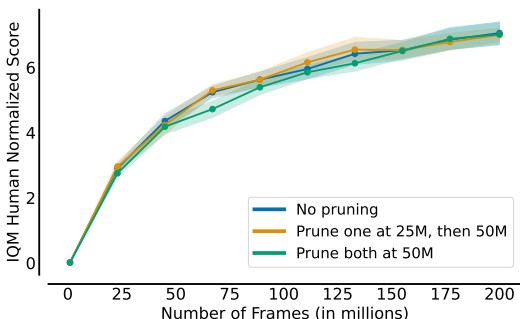

Figure 19: Pruning subset of experts in SoftMoE-4 does not lead to performance drop, suggesting that some of the experts are redundant. We report interquantile mean performance with error bars indicating 95% confidence intervals, over 20 games with 5 seeds each.

### B.5.2 IMPROVING EXPERT UTILIZATION

**Experimental details** For hyper-parameter tuning, we used six games (Asterix, Demon Attack, Seaquest, Breakout, Beam Rider, Space Invaders). For the reset period, we searched for Rainbow over the grid $[0.5 \times 10^4, 10 \times 10^4, 50 \times 10^4, 100 \times 10^4, 125 \times 10^4, 500 \times 10^4, 1000 \times 10^4]$ gradient steps. The best found is $125 \times 10^4$, which corresponds to 20M environment steps. For S&P, we searched over the values $[(0.5, 0.5), (0.4, 0.1), (0.8, 0.2)]$ for shrink and perturb hyperparameters respectively, as studied in (Schwarzer* et al., 2023). We used values of (0.5, 0.5).

**Reset experts and router weights** In investigating current techniques to improve expert utilization, we explore reset either experts weights only or both experts and router weights. As illustrated in Figure 20, both methods do not help in improving utilization in Rainbow and DQN, with resetting the router weights leads to worse performance.

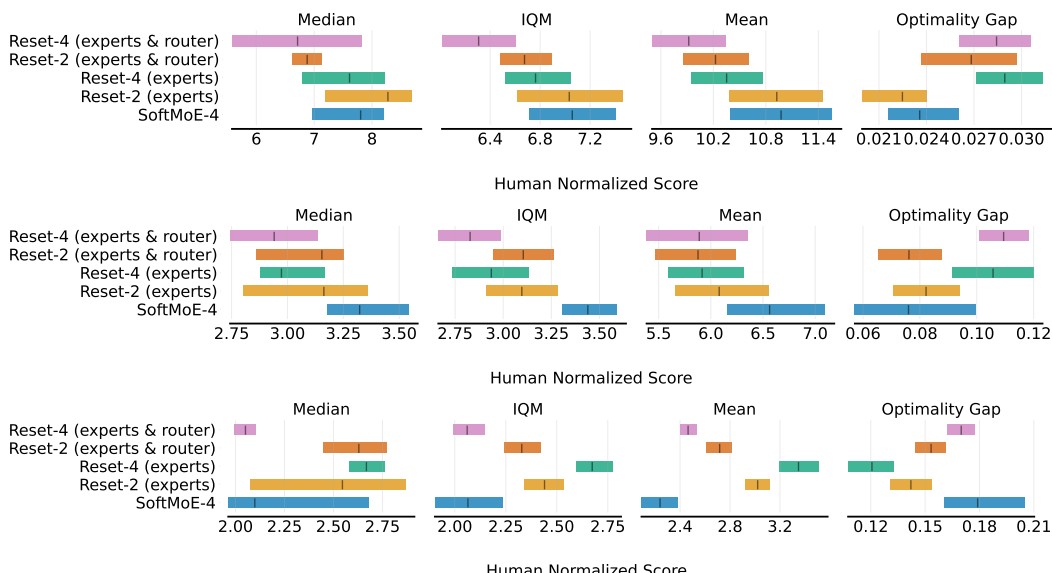

Figure 20: Comparison between resetting experts weights only versus experts and router weights in Rainbow (top), DQN (middle), and DER (bottom).

## C  EXTRA ENVIRONMENTS

To evaluate the generality of our claims, we conducted experiments with Rainbow on Procgen (Cobbe et al., 2019) (Figure 21) and with SAC (Haarnoja et al., 2018) on CALE (Farebrother & Castro, 2024).

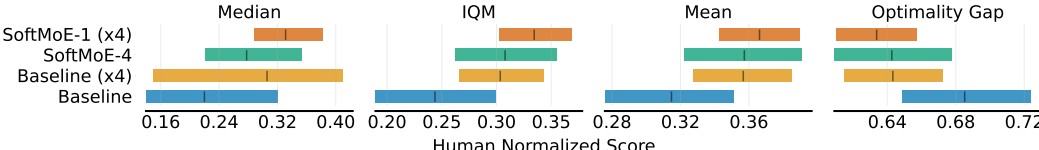

Figure 21: **Evaluating Rainbow on the 16 easy Procgen (Cobbe et al., 2019) environments** for SoftMoE-4 and SoftMoE-1 with the penultimate layer scaled by 4x. Normalization was done using the $R_{min}$ and $R_{max}$ scores from Cobbe et al. (2019). Reporting Median, IQM, Mean, and Optimality Gap (Agarwal et al., 2021), where higher is better for the first three. Consistent with the results in the paper, SoftMoe yields improvements, and the scaled SoftMoE-1 is comparable to SoftMoE-4.

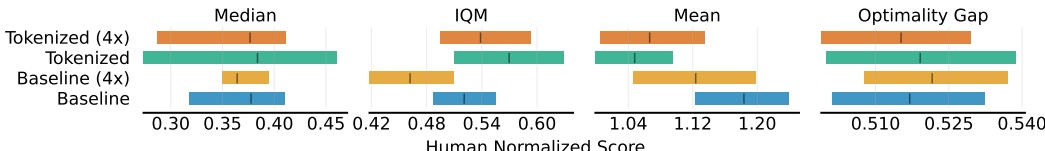

Figure 22: **Evaluating tokenization on the SAC encoder (Haarnoja et al., 2018) on the CALE (Farebrother & Castro, 2024)**, as done in Figure 5. In these experiments, we summed over the first dimension after tokenization. Reporting Median, IQM, Mean, and Optimality Gap (Agarwal et al., 2021), where higher is better for the first three. Consistent with the results in the paper, tokenization yields improvements over the baseline.

## C.1 PER-GAME RESULTS

### C.1.1 DON'T FLATTEN, TOKENIZE!

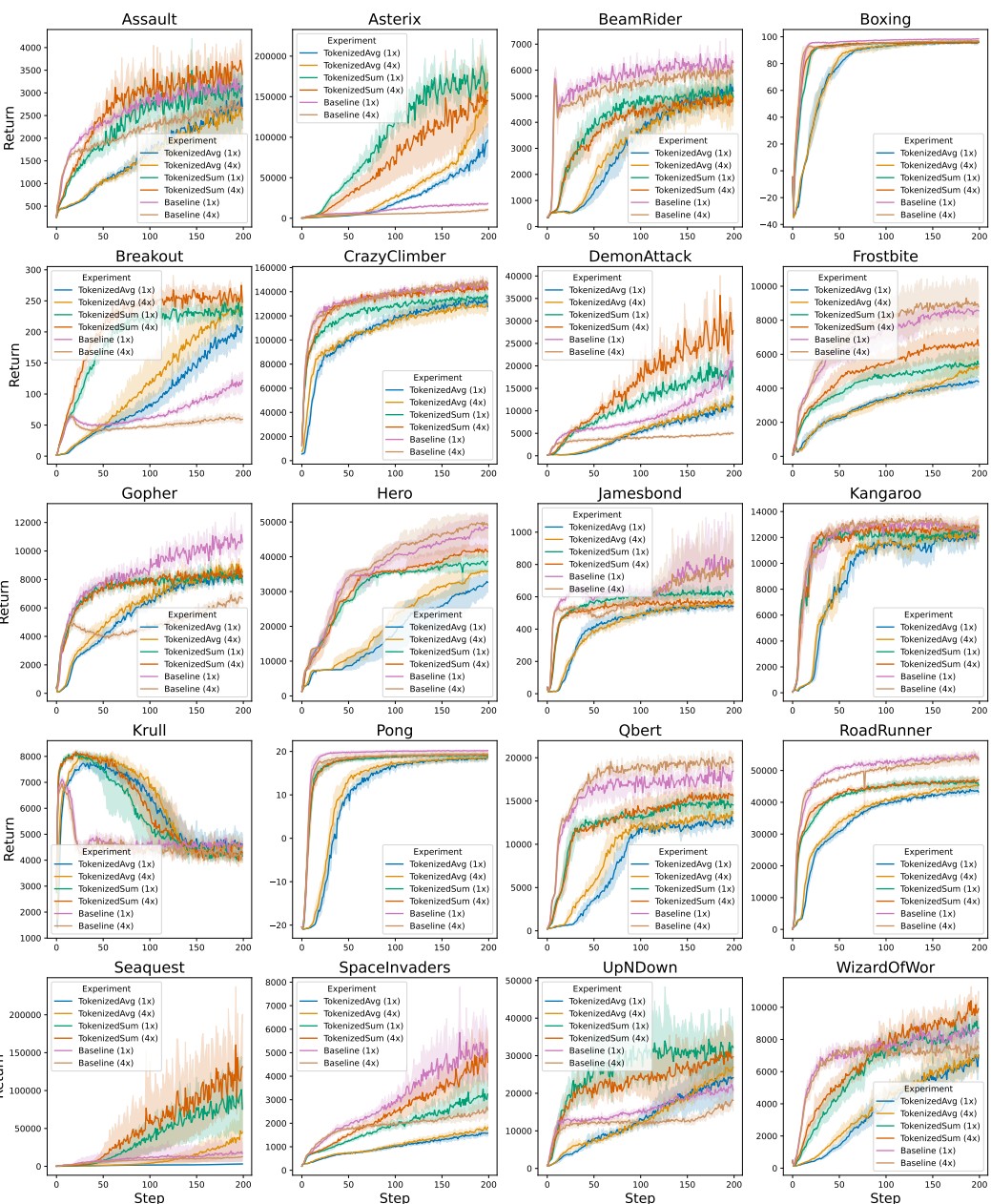

Figure 23: Per-game results for tokenized Rainbow-lite with CNN network (corresponding to Figure 5).

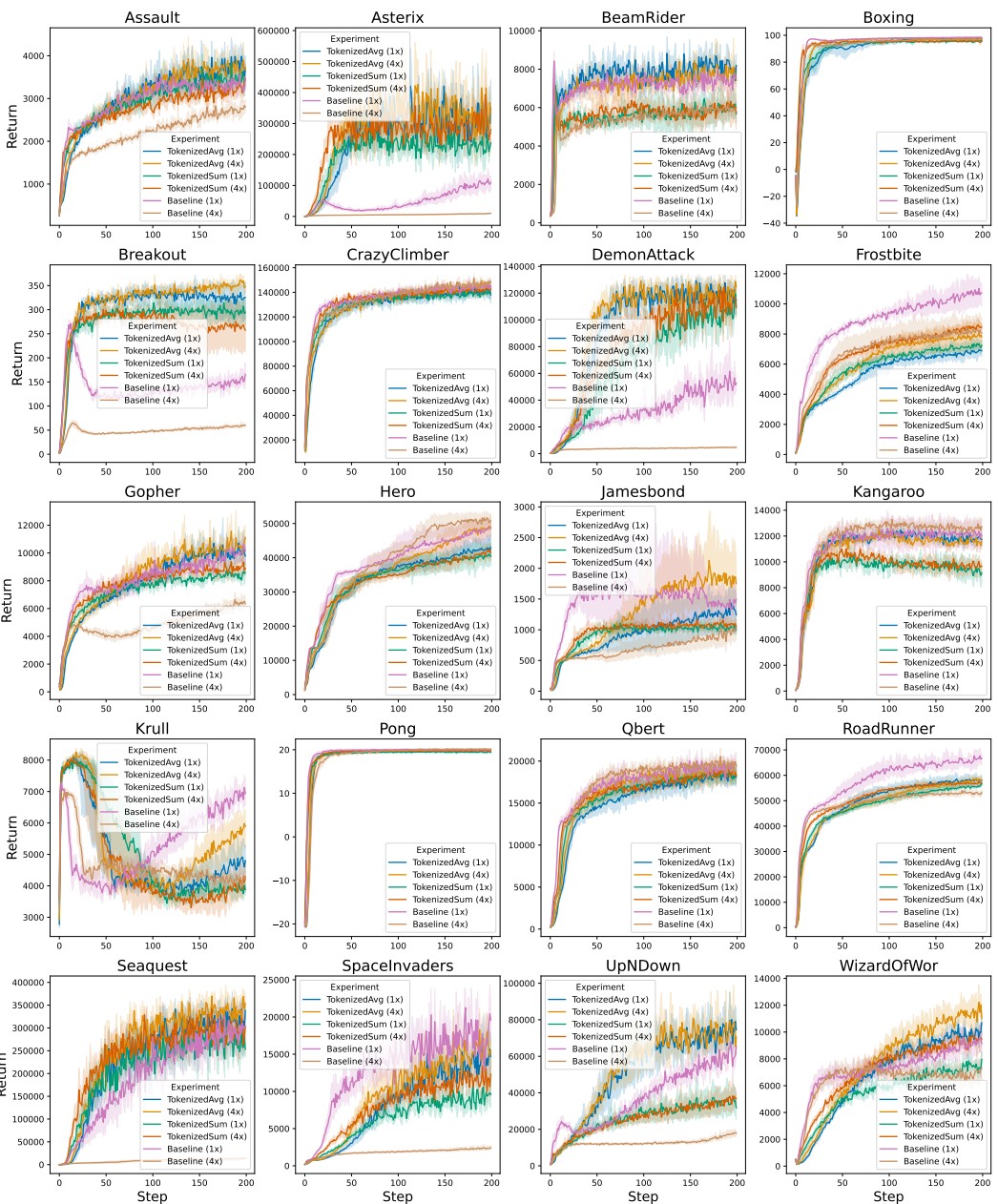

Figure 24: Per-game results for tokenized Rainbow-lite with Impala network (corresponding to Figure 5).

