# OpenReview forum: "Don't flatten, tokenize! Unlocking the key to SoftMoE's efficacy in deep RL"
_ICLR.cc/2025/Conference — ICLR 2025 Spotlight_

### Official Review · Reviewer_XvpH · 2024-11-02

**Soundness:** 2
**Presentation:** 4
**Contribution:** 2
**Rating:** 5
**Confidence:** 4

**Summary:**

MoE (Mixture of Experts) has demonstrated significant potential. The latest research on MoE, exemplified by softMoE, attributes the performance advantages of these algorithms to structural sparsity. However, the authors of this paper propose a contrasting view. They observe that even a single-expert version of softMoE can perform quite well. Additionally, they find no evidence of expert specialization occurring in softMoE.

Through ablation studies, the authors discover that "tokenization" of the output from the convolutional encoder plays a crucial role. As a result, they suggest that tokenization is a highly important operation.

The authors investigate the importance of different components of softMoE in Atari game experiments, including: (i) the use of a learnable tensor Φ for obtaining dispatch (ΦD) and combine (ΦC) weights; (ii) processing p input slots per expert; (iii) architectural dimensions (network depth and width); (iv) the use of n experts; and (v) the tokenization of the encoder output.

A key finding is that tokenization significantly enhances performance. By applying only tokenization to the baseline, a notable performance improvement is observed, demonstrating that tokenization is a primary contributor to the effectiveness of softMoE. Furthermore, even a single-expert softMoE achieves relatively strong results.

Finally, the authors explore methods to improve the performance of multi-expert MoE models. They attempt reset and S&P methods, testing them with Rainbow and DER algorithms, but do not find any approach that consistently enhances multi-expert MoE performance.

**Strengths:**

The authors clearly present three different classical methods for processing encoder features: Flatten (baseline), SoftMoE architecture, and the routing mechanism.

**Weaknesses:**

1) The paper has a significant issue: the baseline scaled by *4 used throughout the experiments appears to perform worse, potentially due to suboptimal parameter. The use of the *4 baseline is problematic, as it may unfairly weaken the baseline, thus exaggerating the benefits of tokenization.

The authors could run experiments comparing unscaled baselines to SoftMoE models with equivalent parameter counts. Specifically, It is crucial to compare the original-sized baseline with the SoftMoE where experts are scaled down by four times, and also compare it with a single expert SoftMoE scaled down by four times.

When examining Figure 5, comparisons between unscaled results reveal that both the tokenized sum and tokenized average actually perform worse. This suggests that the effectiveness of tokenization depends on a balance between network structure, task complexity, and parameter settings, rather than indicating that tokenization is inherently beneficial.

2) The paper also mentions that a single-expert SoftMoE performs well, which suggests that multiple experts may not be critical, and that tokenization is the key factor. However, Figure 10 indicates that combining a single expert with tokenization in SoftMoE does not improve performance in DQN. This contradict the claims that a single expert is effective. This indicates that the paper’s main assertions are not solid. Could the authors address this discrepancy and provide additional analysis or experiments to clarify the effectiveness of single-expert SoftMoE across different algorithms？

The truth might be that the authors' DQN setup is relatively underperforming, making the multiple-expert configuration beneficial to compensate the weakness of single expert. In contrast, stronger algorithms perform well enough with a single expert, making multiple experts redundant. Testing on more complex tasks might reveal a clearer advantage of using multiple experts over a single one. Therefore, a more reasonable interpretation would be that both tokenization and multiple experts contribute to SoftMoE’s performance, aligning with the fundamental concept of MoE rather than overemphasizing the importance of tokenization.

3) In Section 6, the authors aim to develop new techniques to improve expert utilization and maximize the benefits of MoE architectures in RL. However, only two tricks are listed, and the number of baselines compared is too limited. The paper fails to present methods that consistently improve multi-expert MoE performance. If the goal is to provide which trick may improve certain algorithm, a broader set of techniques should be explored.

**Questions:**

1) The authors mentioned that "they demonstrated that a performance drop is not observed when scaling down the dimensionality of each expert." Did you experiment with a setup where the dimensionality is reduced by 4 times in the SoftMoE model? Specifically, the experimental group would use SoftMoE with 4 experts, each scaled down by a factor of 4 to maintain the same overall model size as the baseline. The control group is baseline, which would not be scaled. This setup might highlight the importance of the number of experts in SoftMoE if the baseline performance remains unaffected.

2) Could the authors provided training curves for each individual Atari game? Comparing the performance of the Unscaled baseline and Unscaled tokenize models on each task.

3) Did the authors conduct experiments on environments like DeepMind Control (DMC) or Meta-World to thoroughly demonstrate the generalization capabilities of the tokenize?

4) Why is "Network Plasticity in RL" discussed in the related work section? Which part of the paper is relevant to plasticity? Does it correspond to Section 6? If so, the experimental results presented in Section 6 do not seem to lead to any valuable conclusions.

---

> ### Author Response · Authors · 2024-11-20
> **Rebuttal by Authors (1/2)**
>
> We thank the reviewer for their feedback, useful comments, and address their concerns below.
>
> We appreciate the opportunity to provide a clarification regarding the reviewer’s summary. Our work does not aim to provide “a contrasting view” of recent MoE research, nor are we trying to suggest that a single expert suffices. Rather, our analyses reveal that multiple experts are not being fully utilized in the specific context of online deep RL settings studied in [1]. This finding has an important implication on encouraging further research to increase their usage for further performance improvement as discussed in Section 1 and Section 8 and also acknowledged by Reviewers gzLL and 3aV6. With an awareness of the potential benefits of multiple experts, our work has already started to explore techniques from existing literature to enhance network utilization, as presented in Section 6, though this is not the paper's primary focus.
>
> > **W1:** The paper has a significant issue: the baseline scaled by *4 used throughout the experiments appears to perform worse, potentially due to suboptimal parameter. The use of the *4 baseline is problematic, as it may unfairly weaken the baseline, thus exaggerating the benefits of tokenization.
>
> **A:** Given that our work stems from the ideas explored in [1], we used the same scaled baselines used there. In contrast to [1], we studied the effect of the upscaling of each expert and found that agents do not experience the performance drop observed in the scaled baseline. Further details can be found in Section 4.2.
>
> > **W2:** Could the authors address this discrepancy and provide additional analysis or experiments to clarify the effectiveness of single-expert SoftMoE across different algorithms?
>
> **A:** Indeed, as discussed in section 5.1 (and in [1]), DQN appears to benefit less from the use of SoftMoEs in general, which may explain why SoftMoE-1 yields little gain. We hypothesize this may be due DQN’s use of regression versus Rainbow’s classification (C51) loss; we are currently running experiments with SoftMoE-1 (x4) with the C51 loss and will report back here once they have made more progress.
>
> > **W2:** a more reasonable interpretation would be that both tokenization and multiple experts contribute to SoftMoE’s performance
>
> **A:** Certainly, multiple experts do contribute to SoftMoE’s performance, as demonstrated by the higher performance of multiple experts compared to a single scaled expert in our figures. However, the relatively small performance difference between these two, compared to the larger difference between the scaled baseline and the single scaled expert, suggests that multiple experts are not the primary driver of the observed performance improvement; more importantly, it suggests that we are under-utilizing multiple experts, as we discussed in Section 6.
>
> > **W3:** In Section 6, the authors aim to develop new techniques to improve expert utilization. However, only two tricks are listed, and the number of baselines compared is too limited. If the goal is to provide which trick may improve certain algorithm, a broader set of techniques should be explored.
>
> **A:** As we clarified, the paper's main focus is to understand the reasons behind the observed efficiency of SoftMoEs in deep RL. In Section 6, we take the first step towards a promising future direction for improving expert utilization by studying existing techniques.

---

> > ### Author Response · Authors · 2024-11-20
> > **Rebuttal by Authors (2/2)**
> >
> > > **W1,Q1:** The authors could run experiments comparing unscaled baselines to SoftMoE models with equivalent parameter counts. Specifically, It is crucial to compare the original-sized baseline with the SoftMoE where experts are scaled down by four times, and also compare it with a single expert SoftMoE scaled down by four times. Did you experiment with a setup where the dimensionality is reduced by 4 times in the SoftMoE model?
> >
> > **A:** As mentioned in Section 4, evaluating down-scaled experts was explored in [1] (section 4.2), where the dimensionality is reduced by 4 times. Note that a single SoftMoE expert scaled down by 4 does not match the parameter count of the unscaled baseline. Instead, [1] shows that with matched hidden dimensionality, SoftMoE-1 outperforms the baseline. We believe that our findings help explain the reason behind this. Nevertheless, based on the reviewer's suggestion, we are currently running the SoftMoE-1 scaled down by 4 and will include the results once available.
> >
> > > **Q2:** Could the authors provided training curves for each individual Atari game? Comparing the performance of the Unscaled baseline and Unscaled tokenize models on each task.
> >
> > **A:** We have added per-game results in Appendix B.6.
> >
> > > **Q3:** Did the authors conduct experiments on environments like DeepMind Control (DMC) or Meta-World to thoroughly demonstrate the generalization capabilities of the tokenize?
> >
> > **A:**  Our work primarily focuses on the same domains examined in [1]. We agree that expanding MoE research in RL to include other domains would be an intriguing direction for future work (as mentioned in the discussion).
> >
> > > **Q4:** Why is "Network Plasticity in RL" discussed in the related work section? ..Does it correspond to Section 6? If so, the experimental results presented in Section 6 do not seem to lead to any valuable conclusions.
> >
> > **A:** Yes, it corresponds to Section 6; the prior works we are leveraging are generally regarded as efforts towards improving network utilization. Our intent with that section, and its related literature, is primarily to indicate a promising avenue for future direction.
> >
> > We hope that our answers help address the reviewer’s concerns. We appreciate the reviewer’s time in reading our explanations!
> >
> > [1] Ceron, Johan Samir Obando, et al. "Mixtures of Experts Unlock Parameter Scaling for Deep RL." Forty-first International Conference on Machine Learning.

---

> ### Comment · Reviewer_XvpH · 2024-11-25
>
> Thank the authors for answering each of my questions in detail. However, my main concerns remain unresolved. Specifically:
>
> 1. **Baseline unscaled** consistently outperforms **Baseline scaled ×4** in multiple metrics (as shown in Fig. 5: CNN and Impala's median, IQM, and mean). Therefore, I believe it is reasonable and necessary to compare against **Baseline unscaled**.
>
> 2. If we compare **Baseline unscaled** with **tokenized sum/avg unscaled**, the conclusions of the paper would not hold. The results suggest that tokenization is not the primary driver of performance improvement. (Refer to Fig 5: CNN and Impala's median, IQM, Mean; Appendix B.6.1  Assault, BeamRider,  Boxing, CrazyClimber, Frostbite,Gopher,Hero...) In 25 environment of total 40 environment,  Baseline unscaled performs better then tockenized sum unscaled and tockenized avg unscaled in the same time, which means **Tokenization is not the primary driver of the performance improvement**.

---

> > ### Author Response · Authors · 2024-11-26
> > **Addressing remaining concerns**
> >
> > Thank you for your response. The results in Figure 5 were meant to demonstrate that tokenization (as opposed to flattening) can yield strong improvements. As suggested by reviewer 8sGc, we ran extra experiments with PerFeat tokenization which show even stronger performance improvements over the unscaled baseline (see updated Figure 5). We believe that these add extra evidence to our paper’s claim on the sub-optimality of flattening.
> >
> > We have included the requested experiment with SoftMoE-1 scaled down by 4 in Appendix B.3. As shown in Figure 17, it still outperforms the baseline despite its penultimate layer having four times fewer parameters.
> >
> > To further verify the generality of our claims, in Appendix C we include experiments run with Rainbow on Procgen [2] (Figure 21) and SAC [3] on the CALE [4] (Figure 22), which yield results consistent with our paper’s claims.
> >
> > Regarding the comparison against an unscaled baseline, our focus was on investigating SoftMoEs efficacy in _scaling_ models, as explored by [1]. Indeed, the main claim of [1] was that SoftMoEs helps avoid the performance collapse when scaling up parameters; our work stems from this finding and argues that the main component for enabling this type of scaling is tokenization. While we agree that the results when tokenizing the unscaled baseline are not as strong, it is important to consider that these are initial experiments without any hyper-parameter (or configuration) optimization.
> >
> > We hope these new results and comments are sufficient to address your concerns.
> >
> > [1] Ceron, Johan Samir Obando, et al. "Mixtures of Experts Unlock Parameter Scaling for Deep RL." Forty-first International Conference on Machine Learning.
> >
> > [2] Karl Cobbe, Christopher Hesse, Jacob Hilton, and John Schulman.  Leveraging procedural generation to benchmark reinforcement learning. arXiv preprint arXiv:1912.01588, 2019.
> >
> > [3] Tuomas Haarnoja, Aurick Zhou, Pieter Abbeel, and Sergey Levine.   Soft actor-critic:  Off-policy maximum entropy deep reinforcement learning with a stochastic actor.  In International conference on machine learning, pp. 1861–1870. PMLR, 2018.
> >
> > [4] Jesse Farebrother and Pablo Samuel Castro.  CALE:  Continuous arcade learning environment. Advances in Neural Information Processing Systems, 2024.

---

> > > ### Author Response · Authors · 2024-11-27
> > > **Any remaining concerns?**
> > >
> > > Dear reviewer,
> > > Given that today is the last day we can update the PDF, we wanted to check in to see if you felt there were still unaddressed concerns and if not, we would invite to reconsider your score.
> > > Thank you!

---

> > > ### Comment · Reviewer_XvpH · 2024-11-27
> > >
> > > Sorry for the delay. You mentioned that "tokenizing the unscaled baseline is not as strong." In fact, it performs worse sometimes, especially for the median and IQM. This is my main concern. I am not sure if there is any misunderstanding here. In my opinion, the comparison between "tokenizing the unscaled baseline" and the "unscaled baseline" is the only fair way to proceed.
> > >
> > > There is no need to worry about the submission deadline for the PDF. I believe we can discuss the results based on the current PDF.

---

> ### Author Response · Authors · 2024-11-27
>
> Thank you for your reply!
>
> As previously mentioned, [1] proposes SoftMoEs that enables scaling RL networks with performance increases. Our primary objective in this paper was to investigate _why_ SoftMoE is so effective at enabling this scaling, which is why most of our experiments have focused on the scaled baseline.
>
> However, we do agree that it is useful to evaluate the impact of tokenization on unscaled models. We have added Figure 16 in the appendix which only compares the (unscaled) baseline against the tokenized counterparts. Even in this case, we do see gains with tokenization, especially when using the CNN architecture. With the Impala architecture we see strong gains with tokenization as measured by the Mean and comparable performance with IQM. We report these 4 metrics as they provide a clearer picture of the difference between the methods (see [2] for an in-depth discussion of their differences). Furthermore, as we observe in Figures 5 and 17 with 4x scaling, PerFeat seems to be a stronger tokenization approach than PerPixel when using without SoftMoE; this suggests that the gains we observe with the unscaled baseline can be made larger with PerFeat tokenization. Finally, in Figure 15 we explored replacing flattening with global average pooling, as suggested by reviewer gzLL, which shows significant gains on both the unscaled and scaled baseline.
>
> We hope this addresses your remaining concern?
>
> [1] Ceron, Johan Samir Obando, et al. "Mixtures of Experts Unlock Parameter Scaling for Deep RL." Forty-first International Conference on Machine Learning.
>
> [2] Rishabh Agarwal, Max Schwarzer, Pablo Samuel Castro, Aaron C Courville, and Marc Bellemare.Deep reinforcement learning at the edge of the statistical precipice.  In M. Ranzato, A. Beygelz-imer, Y. Dauphin, P.S. Liang, and J. Wortman Vaughan
> (eds.),Advances in Neural InformationProcessing Systems, volume 34, pp. 29304–29320. Curran Associates, Inc., 2021.

---

> > ### Comment · Reviewer_XvpH · 2024-11-28
> >
> > Your explanation of Figure 16 has somewhat alleviated my concerns. The Impala mean results do indeed support the author's argument. Unfortunately, the performance is not as evident in the median and IQM metrics. There is no consistent performance improvement in the CNN experiments. This suggests that while some experimental results provide evidence for the author's conclusion, the evidence is not particularly strong.
> >
> > Moreover, the author mentions that the problem they aim to solve is _why SoftMoE is so effective at enabling scaling RL networks_. Although I am skeptical about the significant performance drop in the scaled baseline, the experimental results in the paper do provide support for the problem the author is trying to address.
> >
> > Taking these comments into account, I have raised the score to 5. I encourage the authors to conduct further tuning of the scaled baseline in future work to rigorously ensure the validity of performance drop.

---

> > > ### Author Response · Authors · 2024-12-01
> > >
> > > Thank you for your response, and for willingness to adjust your rating of our paper!
> > > As previously mentioned, this particular experiment in question was meant as an indicator of the sub-optimality of flattening, which is what is most commonly used in the literature.
> > >
> > > Our choice of PerConv tokenization was driven by the results we observed when combined with SoftMoE. However, as prompted by reviewer 8sGc, we repeated this experiment with PerFeat tokenization. As can be seen in [this figure](https://anonymous.4open.science/r/rebut-8353/newFig5.png), PerFeat performs stronger than PerConv when used in isolation and, importantly, consistently outperforms the baseline in both scaled and unscaled variants, in both architectures considered, and across all 4 metrics.
> > >
> > > We agree with your initial suggestion that we should be comparing with the unscaled baseline more directly. For this reason, we will be replacing the current Figure 5 (which is splitting the two figures across network architectures) with the [new linked figure](https://anonymous.4open.science/r/rebut-8353/newFig5.png), which splits unscaled and scaled comparisons ([new version of paper](https://anonymous.4open.science/r/rebut-8353/Don_t_flatten__tokenize.pdf)). This clarifies the main message of the figure: don’t flatten!
> > >
> > > Regarding the scaled baseline, our experiments are consistent with the results of [1], and it is worth noting that the strong performance we see with PerFeat in the new Figure 5 is also without any tuning.
> > >
> > > We thank you for pushing us on this point, as it has made our results (and our paper) stronger. We believe these new results should provide the consistency in gains you were expecting; if so, we would invite you to consider raising your score again, above the acceptance threshold.

---

> > > > ### Author Response · Authors · 2024-12-02
> > > > **Official Comment by Authors**
> > > >
> > > > Dear reviewer, as the discussion period is coming to an end, we wanted to provide a summary of our answers to the main questions/concerns:
> > > >
> > > > - We clarified that our paper's goal is to understand the efficiency of SoftMoE in scaling RL models as highlighted by recent work [1], rather than to give a contrasting view or suggest a single expert is sufficient. Our findings highlight two future directions:
> > > >
> > > >    (1) exploring alternatives to the commonly used flattening operation, and
> > > >
> > > >    (2) improving expert utilization (see Sections 1 and 8 for details).
> > > >
> > > > - We provided experiments with Rainbow on Procgen and SAC on the continuous action version of ALE with results consistent with our main paper’s findings (Appendix C), providing further evidence for the generality of our claims.
> > > >
> > > > - We performed experiments further confirming that replacing the flattening operation consistently leads to performance improvement in the *unscaled* and *scaled* baselines across different architectures ([*new* Figure 5](https://anonymous.4open.science/r/rebut-8353/newFig5.png)), and without the need of any extra tuning.
> > > >
> > > > - We clarified that the requested experiments with scaled down dimensionality of experts have already been studied in [1] (and discussed in Section 4 of our paper), and added a comparison of SoftMoE-1 with a scaled down variant, as suggested (Appendix B.3).
> > > >
> > > > Since today is the last day of discussion, we kindly ask the reviewer to evaluate our response and reconsider their score in light of our clarifications to all raised questions.
> > > >
> > > > [1] Ceron, Johan Samir Obando, et al. "Mixtures of Experts Unlock Parameter Scaling for Deep RL." Forty-first International Conference on Machine Learning.

---

### Official Review · Reviewer_8sGc · 2024-11-03

**Soundness:** 3
**Presentation:** 4
**Contribution:** 3
**Rating:** 8
**Confidence:** 4

**Summary:**

Obando-Ceron* et al. (2024) [1] demonstrated that SoftMoEs are effective architectures for scaling models in online RL. However, their paper did not explain the reasons behind this performance gain at scale. This submission analyzes factors that might contribute to the effectiveness of SoftMoEs at scale by ablating different components of the SoftMoE architecture within the Rainbow, DER, and DQN baselines on the Atari benchmark.

Overall, the authors show that tokenizing the encoder output, rather than using multiple experts, is the primary factor driving SoftMoE’s effectiveness. They also demonstrate that a single scaled expert with tokenization can match the performance of multiple experts.

[1]: Ceron, Johan Samir Obando, et al. "Mixtures of Experts Unlock Parameter Scaling for Deep RL." Forty-first International Conference on Machine Learning.

**Strengths:**

- The paper is well-written, easy to follow, and well-motivated.
- The authors conducted numerous empirical ablations on 3 different algorithms—primarily on Rainbow, but also on DER and DQN—using the popular Atari benchmark.
- I appreciate that the authors reported the IQM and Optimality Gap on 5 seeds for statistical significance.

**Weaknesses:**

While this paper provides an in-depth exploration of tokenization in SoftMoEs, there are areas where it could be improved to make it an accept.

1. The paper’s scope feels somewhat narrow, as it primarily focuses on *why* SoftMoEs work well when scaled up on the Atari benchmark using the Rainbow, DER, or DQN algorithms alone. As a result, it remains unclear whether SoftMoEs would be effective in more challenging OOD generalization benchmarks, such as Procgen, which present greater difficulty than the IID singleton environments like Atari.
2. Additionally, as shown in Figure 10, the SoftMoE-1 (scaled 4x) baseline performs significantly worse in the DQN setting. It would be beneficial for the authors to test their scaled-single-expert approach on another algorithm, potentially an actor-critic algorithm like PPO, in a pixel-based environment like Procgen.
3. Given the popularity of DQN, further investigation into the differences in DQN’s behavior compared to the other two algorithms would also add more value to this submission.

**Questions:**

I have some questions based on the current state of submission:

- For all trends observed in Section 4.2, do the authors anticipate that similar trends would hold for DER? Could the hypotheses be verified on an additional algorithm to confirm that the design choices assessed in Section 4.2 are generally applicable and not specific to one algorithm i.e. Rainbow?
- In Figure 5, did the authors observe a clear scaling trend from 1x to 2x, 4x, and 8x using the tokenized_sum baseline with either CNN or IMPALA? Also, in Figure 5, the tokenized scheme was [h\*w, d], which I believe corresponds to PerConv tokenization. Would similar trends hold if [d, h\*w] (i.e., PerFeat tokenization, similar to Figure 7) were used instead? This would help confirm whether tokenization generally improves performance, even if PerConv outperforms PerFeat, as long as both do better than simple flattening on the baseline.
- In Figure 8, when selecting 10% of the slots, are these slots chosen randomly, or is a specific heuristic used for pruning? Additionally, are these 10% slots fixed throughout training, or do they change randomly at each training iteration?
- Why was the analysis in Section 6 (Figure 11) not conducted for DQN?


**Typos:**

Page 3 Line 134: propose —> proposed

Page 8 Line 385: performance on all the 60 games —> performance on all **of** the 60 games

---

> ### Author Response · Authors · 2024-11-20
> **Rebuttal by Authors**
>
> We thank the reviewer for their feedback! We are glad that the reviewer found the “analysis in-depth” and the “paper is well-written”.
>
> > **W1, W2:** primarily focuses on why SoftMoEs work well when scaled up…As a result, it remains unclear whether SoftMoEs would be effective in more challenging OOD generalization benchmarks, such as Procgen, which present greater difficulty than the IID singleton environments like Atari. … It would be beneficial for the authors to test their scaled-single-expert approach on another algorithm, potentially an actor-critic algorithm like PPO, in a pixel-based environment like Procgen.
>
> **A:** As the reviewer mentioned, we focus in this work on understanding the significant performance improvement observed by prior work [1] on the studied benchmarks. Although we are running on single-task settings, there is a high-degree of non-stationarity due to the evolving policy. Our findings have implications on future research mainly by 1. revisiting the architectural choices and replacing the flattening operation, and 2. improving expert utilization in MoEs for further performance improvements. We think that studying the effectiveness of SoftMoEs in OOD generalization is an orthogonal, *but* valuable future line of research. We are currently setting up infrastructure to evaluate PPO on ProcGen, as suggested by the reviewer, and will report back here when we have results.
>
> > **W3:** Given the popularity of DQN, further investigation into the differences in DQN’s behavior compared to the other two algorithms would also add more value to this submission.
>
> **A:** Indeed, as discussed in section 5.1 (and in [1]), DQN appears to benefit less from the use of SoftMoEs in general, which may explain why SoftMoE-1 yields little gain. We hypothesize this may be due DQN’s use of regression versus Rainbow’s classification (C51) loss; we are currently running experiments with SoftMoE-1 (x4) with the C51 loss and will report back here once they have made more progress.
>
> >**Q1:** For all trends observed in Section 4.2, do the authors anticipate that similar trends would hold for DER? “Could the hypotheses be verified on an additional algorithm to confirm that the design choices assessed in Section 4.2 are generally applicable and not specific to one algorithm i.e. Rainbow?”
>
> **A:** We thank the reviewer for their nice suggestion! We have run the full analyses in Section 4.2 on DER and added the results in Appendix B.3 and pointed to it in Section 4.2. We find that all the observations are consistent with our previous results as shown in Figure 16.
>
> > **Q2:** In Figure 5, did the authors observe a clear scaling trend from 1x to 2x, 4x, and 8x using the tokenized_sum baseline with either CNN or IMPALA? Also, in Figure 5, the tokenized scheme was [h*w, d], which I believe corresponds to PerConv tokenization. Would similar trends hold if [d, h*w] (i.e., PerFeat tokenization, similar to Figure 7) were used instead? This would help confirm whether tokenization generally improves performance, even if PerConv outperforms PerFeat, as long as both do better than simple flattening on the baseline.
>
> **A:** We thank the reviewer for this suggestion. We are currently running experiments with tokenization and 2x scaling, to verify whether we observe a scaling trend. With regards to tokenization, we selected PerConv tokenization for Figure 5 given that we found it to be the most performant (see Figure 7). However, we are currently running experiments with PerFeat tokenization to see if the same trend holds, as suggested by the reviewer. We will report back once the experiments are completed.
>
> > **Q3:** In Figure 8, when selecting 10% of the slots, are these slots chosen randomly, or is a specific heuristic used for pruning? Additionally, are these 10% slots fixed throughout training, or do they change randomly at each training iteration?
>
> **A:** We are not selecting 10% of the slots. The number of slots p is a predefined constant number which determines the capacity of each expert and it is typically fixed throughout training. In this experiment, we just reduced the default value of this number by 90%. We refer the reviewer to the added description to Section 3 regarding the choice of $p$ (number of slots), and have added more details to section 5.1 to avoid confusion.
>
> > **Q4:** Why was the analysis in Section 6 (Figure 11) not conducted for DQN?
>
> **A:** Similar to the rest of the paper, we focused on the cases where SoftMoE was observed to achieve significant performance gains over the baseline. Following your suggestion, we are running some analysis on DQN and will include the results in Appendix B.5.
>
> [1] Ceron, Johan Samir Obando, et al. "Mixtures of Experts Unlock Parameter Scaling for Deep RL." Forty-first International Conference on Machine Learning.

---

> > ### Comment · Reviewer_8sGc · 2024-11-24
> > **Thanks for the rebuttal**
> >
> > I thank the authors for their rebuttal and for running additional experiments to improve the quality of this submission. The responses addressed most of my concerns and I have raised my rating to 8.

---

### Official Review · Reviewer_gzLL · 2024-11-06

**Soundness:** 3
**Presentation:** 2
**Contribution:** 3
**Rating:** 8
**Confidence:** 3

**Summary:**

This paper analyzes the effect of different components in the soft-of-mixture of experts (SoftMoEs) in online RL, the goal is to understand the key factors and design decisions that influence/derive the performance of (SoftMoEs), the analysis shows that tokenizing the output of the cnn encoder has the biggest effect on the performance, even when using a single expert or the baseline model, tokenizing the encoder output results.

**Strengths:**

- Results are significant with small variance over the 60 Atari games.

- The effect of tokenization seems to transfer between architectures, which might suggest that using tokenization would always be helpful, at least in the algorithms used in the paper (Rainbow and DER).

- Showing that we are underutilizing the mixture of experts in (SoftMoEs) is an important implication of this paper, which encourages more research in this area.

**Weaknesses:**

- Authors should add a section that explains what a token is and what a slot it is, this is not defined in the paper and it will make the paper much more clearer.

- The effect of tokenization in a single expert does not seem to transfer to DQN, which suggests there is something missing in the analysis, the authors suggested that it might be related to the categorical loss used in Rainbow and DER, but there is no further investigation.

**Questions:**

- In line 361, computational efficiency by combined tokens, I do not understand the point of the plot, is it to show that using fewer slots (which means better time complexity) still results in good performance? Can you add a plot that directly shows the relationship between the number of slots and time complexity?

- The authors argue that tokenizing the encoder output preserves the spatial information unlike flattening, can the authors run a baseline where the cnn encoder output is actually a vector? This can be done by adding a global average pooling layer after the last conv layer, which will reduce the spatial dimension to one.

---

> ### Author Response · Authors · 2024-11-20
> **Rebuttal by Authors**
>
> We thank the reviewer for their feedback! We are pleased to hear that they find the “results significant” and that our finding on expert underutilization has “an important implication, encouraging more research in this area”.
>
> > **W1:** Authors should add a section that explains what a token is and what a slot it is, this is not defined in the paper and it will make the paper much more clearer.
>
> **A:** Thank you for your suggestion, we have added more clarification about the slot in the revised version, in addition to what is presented at the end of section 3 (and in Figure 3).
>
> > **W2:** DQN … further investigation
>
> **A:** Indeed, as discussed in section 5.1 (and in [1]), DQN appears to benefit less from the use of SoftMoEs in general, which may explain why SoftMoE-1 yields little gain. We hypothesize this may be due DQN’s use of regression versus Rainbow’s classification (C51) loss; we are currently running experiments with SoftMoE-1 (x4) with the C51 loss and will report back here once they have made more progress.
>
> > **Q1:** is it to show that using fewer slots (which means better time complexity) still results in good performance? Can you add a plot that directly shows the relationship between the number of slots and time complexity?
>
> **A:** Your interpretation is correct! Another benefit of combined tokens is reducing the computational time without performance drop. We can achieve almost the same results using only 10% of expert slots if slots contain combined tokens unlike the case of sparse tokens. Using 10% of the slots means that the number of processed inputs of each expert is reduced by 90%. We have compared the wall-time in the two cases and observed time saving, but it is relatively marginal given the size of networks we are using.
>
> > **Q2:** can the authors run a baseline where the cnn encoder output is actually a vector?
>
> **A:** Thank you for this insightful suggestion. We have conducted the requested experiments using global average pooling in the default and scaled baselines and included the results in Appendix B.2. Interestingly, consistent with our findings, replacing the flattening operation with global average pooling leads to performance gains in all cases. We appreciate the reviewer’s question which further strengthens the suggestion for revisiting the common practice of flattening the outputs of the convolutional encoders.
>
> [1] Ceron, Johan Samir Obando, et al. "Mixtures of Experts Unlock Parameter Scaling for Deep RL." Forty-first International Conference on Machine Learning.

---

### Official Review · Reviewer_9z4X · 2024-11-06

**Soundness:** 3
**Presentation:** 3
**Contribution:** 3
**Rating:** 8
**Confidence:** 3

**Summary:**

This paper investigates the key factors that make SoFeMoE effective in visual reinforcement learning. Through comparisons, it demonstrates that tokenization and the combination of token weights, rather than simply using multiple experts, drive the performance improvements. The conclusions are based on experiments conducted in the Arcade Learning Environment, covering 60 games and using 5 seeds.

**Strengths:**

- The paper is well-written and easy to read.
 - The figures effectively illustrate the different analyses and help understand the main results.
 - The analysis is solid and well presented.

**Weaknesses:**

The main concern is the generality of the claim.
 - The experiments are conducted in one simulation platform with all discrete actions. It is unclear whether the same observation is universal applicable to other simulation platforms, especially involving agents with continuous states and actions.
 - All the results are conducted with 4 experts. Would the conclusions be applicable to different numbers of experts?
 - In Figure 6, it will be beneficial to visualize the performance with one unscaled expert to understand the results better.

**Questions:**

Please refer to the weakness section.

---

> ### Author Response · Authors · 2024-11-20
> **Rebuttal by Authors**
>
> We thank the reviewer for their feedback! We are glad that the reviewer found the “analysis solid” and the paper is “well-presented”.
>
> > **W1:** unclear whether the same observation is universal applicable to other simulation platforms
>
> **A:** Interesting question! In this work, our primary focus is to understand the underlying reasons for the significant performance improvements observed by [1] on discrete tasks. Our findings turn out to help explain why MoE does not yield performance gains in Mujoco environments, as discussed in Section 8. We agree with the reviewer that extending this work by investigating other domains would be a valuable future direction. We have included this in the discussion section in the revised version.
>
> > **W2:** Would the conclusions be applicable to different numbers of experts?
>
> **A:** We did provide the same analyses on 8 experts in the appendix. As presented in Appendix B.2, the findings and conclusions are consistent with the case of 4 experts. Following your suggestion, we have also included the results for 2 experts and observed similar trends. We added a reference to this appendix in Section 5 of the revised version.
>
> > **W3:** In Figure 6, it will be beneficial to visualize the performance with one unscaled expert to understand the results better.
>
> **A:** We have added this to Figure 6, thank you for the suggestion!
>
> [1] Ceron, Johan Samir Obando, et al. "Mixtures of Experts Unlock Parameter Scaling for Deep RL." Forty-first International Conference on Machine Learning.

---

### Official Review · Reviewer_3aV6 · 2024-11-09

**Soundness:** 3
**Presentation:** 3
**Contribution:** 4
**Rating:** 8
**Confidence:** 4

**Summary:**

The paper extensively studies the degree of contributions by each factor of SoftMoEs, a method mitigating the inverse proportionality between the performance and the architecture size of the online value-based deep reinforcement learning (DRL) methods. Despite the effectiveness of the approach, which component of SoftMoEs drives the improvement remains a mystery. The paper discovered that tokenization, the scheme of converting extracted features while maintaining its spatial structure, plays a significant role in SoftMoEs, bestowing the ability for the algorithm with a large network to perform well. From this insight, the paper further extends the argument to the redundancy of the experts, claiming that a single expert performs competitively against the multi-expert under certain conditions.

**Strengths:**

The paper tackles the important problem: *which factors of SoftMoEs benefit the most in mitigating the performance degradation of DRL algorithms with the increase of network size?* The problem is significantly important as it falls into the category of questions that ask the fundamental mechanism of the algorithms. Questions asking the fundamental mechanism of “why” the existing approach is effective are often overlooked in the community but also broadly impact the community in multiple ways (provides the common ground for the future methodologies, encourages the community to rethink the current approaches, etc.). Successfully answering the fundamental question thus turns this paper into an invaluable work with a potentially significant impact on the community.

The main finding of the paper is that tokenization of the extracted features of observations majorly contributes to performance improvements. This argument has been drawn and supported by the series of experiments along with the rigorous empirical analysis (adoption of IQM, 95% confidence interval over the stratified bootstrap samples), which reinforces the legitimacy of the claim.

**Weaknesses:**

- The significance of tokenization is limited to particular settings and does not apply robustly over the value-based online DRL. In fact, Figure 10 depicts that the scaled SoftMoE-1 resembles a similar performance/optimality gap from that of the scaled Baseline on DQN, indicating the minor, near-zero effect on the performance improvement by the tokenization.
- The paper lacks some crucial details on the experimental settings of SoftMoEs. One of the prominent ones is the number of slots $p$. Unknown $p$ value reduces the confidence in the conclusion drawn in paragraph **Expert specialization** (line 231). Here, the paper claims that the specialization of experts is not the primary factor contributing to the high performance by increasing the number of $p$ to the number of tokens. However, the analysis might not be valid if the default value of $p$ is close to the number of tokens, and there is no way the readers can notice this unless specified in the paper. The paper would benefit from clarifying the actual values of the default configurations of SoftMoEs.
- Some ablation studies omit important explanations, making it hard to follow the arguments. For instance, in the paragraph **Tokenization baseline**, the paper replaces the feature flattening operation of the Baseline with a tokenize-and-aggregate (either by average or sum) operation. The results suggest that tokenization with scaled representation significantly improves the performance of the Baseline, Rainbow-lite architecture. However, it is hard to connect this finding to the claim that “tokenization plays a major role in the efficacy of SoftMoE” (line 295). An additional explanation bridging similar logical gaps in the ablation studies (especially from the results to the final statement) would gradually improve the clarity of the paper.
- The effort towards the hyperparameter sweeps is not explicitly mentioned in the paper. In empirical studies, hyperparameter sweeps are necessary for fair comparison, especially when the purpose of comparison is to determine the effectiveness of approaches. Omitting this step weakens the arguments made in section 6, mitigating experts’ redundancy by parameter reset and S&P.
- Although mentioning how to improve expert utilization (section 6) marks an interesting future direction, it also feels unnecessary. This is mainly due to the open-ended analysis and the fact that the argument is slightly out of the main focus of the paper. The consistency and logical flow of the paper would be improved by moving section 6 to the appendix.

In addition to these points, some minor formatting errors and ambiguous presentations caught my attention:

- Some references miss the year of publication. For instance, line 469 contains two works cited without the publication years.
- Inconsistent citation formats. Capitalization of titles, format of venues, etc.
- Some figures lack explanations. Specifically, the target quantity is unclear in the bar plots: Figure 1, 7, and 9. While it is mentioned that a human-normalized score is the target of measurements in the main text (section 4.1), it would be convenient to indicate within the figure. Also, since other figures indicate human-normalized scores, clarifying the target quantity in all the figures would further improve consistency.

**Questions:**

- How does the result depicted in Figure 5 reduce to the claim: “providing strong evidence that tokenization plays a major role in the efficacy of SoftMoE” (line 295)?
- As mentioned in the weaknesses section, some empirically supported claims are skeptical due to the lack of information about the default configurations of SoftMoEs. Would the authors be able to list the precise configurations?
- Is a hyperparameter search for random resets and S&P conducted in section 6?

---

> ### Author Response · Authors · 2024-11-20
> **Rebuttal by Authors**
>
> We thank the reviewer for their feedback! We are happy that the reviewer found the paper has a “potentially significant impact”. It is great to hear that it could be “a common ground for future methodologies”.
>
> >**W1:** The scaled SoftMoE-1 resembles a similar performance/optimality gap from that of the scaled Baseline on DQN
>
> **A:** Indeed, as discussed in section 5.1 (and in [1]), DQN appears to benefit less from the use of SoftMoEs in general, which may explain why SoftMoE-1 yields little gain. We hypothesize this may be due DQN’s use of regression versus Rainbow’s classification (C51) loss; we are currently running experiments with SoftMoE-1 (x4) with the C51 loss and will report back here once they have made more progress.
>
> >**W3, Q1:** Tokenization baseline: …How does the result depicted in Figure 5 reduce to the claim: “providing strong evidence..”
>
> **A:** The only architectural change in experiment in Figure 5 was replacing the flattening operation with tokenization, allowing us to directly assess the impact of tokenization on enhancing the performance of scaled networks. Thanks for noting this, we clarified this sentence in the revised version by rephrasing it to “… plays a major role in the successful scaling of DRL networks”.
>
> >**W2, Q2:** Expert specialization (line 231): Unknown p value ...if the default value of p is close to the number of tokens...Would the authors be able to list the precise configurations?
>
> **A:** Following the common practice in the MoE literature [2,3], we set $p$ to be the total number of tokens divided by the number of experts (unless otherwise specified). This results in a $p$ value that is  $\frac{1}{numexperts}$ times smaller than having $p$ equal to the number of tokens, as discussed in the expert specialization section. Following your suggestion, we have clarified this in Section 3 in the revised version.
>
> >**W4,Q3:** Is a hyperparameter search for random resets and S&P conducted in section 6?
>
> **A:** Yes, we searched for the reset period and the S&P values. We have included these details in the revised version in Appendix B.4 and added a link to it in Section 6.
>
> >**W5:** expert utilization (section 6) marks an interesting future direction….the paper would be improved by moving section 6 to the appendix
>
> **A:** We thank the reviewer for their suggestion. We find it interesting to add a discussion on future work in the main paper, with positive empirical evidence. However, we will keep this suggestion in mind for the camera-ready version, if space becomes an issue.
>
> **W6-8:** We thank the reviewer for providing formatting suggestions. We incorporated your comments in the revised version.
>
> [1] Ceron, Johan Samir Obando, et al. "Mixtures of Experts Unlock Parameter Scaling for Deep RL." Forty-first International Conference on Machine Learning.
>
> [2] Riquelme, Carlos, et al. "Scaling vision with sparse mixture of experts." Advances in Neural Information Processing Systems 34 (2021): 8583-8595.
>
> [3] Gale, Trevor, et al. "Megablocks: Efficient sparse training with mixture-of-experts." Proceedings of Machine Learning and Systems 5 (2023): 288-304.

---

> > ### Comment · Reviewer_3aV6 · 2024-11-24
> > **Thank you for the rebuttal.**
> >
> > I would like to sincerely thank the authors for their efforts in disclosing further details. **Given the authors’ response, I decided to raise my score from 6 to 8**. The precise reasoning follows below:
> >
> > > Indeed, as discussed … have made more progress.
> >
> > Although the finding (the contribution of tokenization) does not necessarily apply to all the value-based online DRL methods, the paper still provides solid evidence to support that the claim applies to the value-based online DRL approaches with regression loss. Including the results of SoftMoE-1 (x4) with C51 loss would further specify the scope of the paper, which will enhance the clarity of the paper.
> >
> > > The only architectural change in an experiment in Figure 5 … successful scaling of DRL networks.
> >
> > The clarification that the authors made in the rebuttal and the paper effectively improves the logical flow connecting the insights from the experiment to the concluding statements. Especially the adjustment of the conclusion from “the efficacy of SoftMoE” to the “successful scaling of DRL networks” effectively improves the connection between the reported results and their implications. This contributes to a stronger emphasis on the main takeaway of the paper.
> >
> > > Following the common practice in MoE literature … we have clarified this in Section 3 in the revised version.
> >
> > Given that the number of slots $p$ is $\frac{1}{numexperts}$ times the number of tokens and the authors examining their claim with four experts, I now find the insignificance of the expert specialization as a valid claim, resolving my initial concern in the review.
> >
> > > Yes, we searched for the reset period and S&P values. … added a link to it in Section 6.
> >
> > The hyperparameters searched for both algorithms cover a sufficient range of values. The authors also clarified which hyperparameter values are used for the reported results. This supplementary information improves the credibility of the results and arguments in section 6.
> >
> > > We thank the reviewer for their suggestion. … for the camera-ready version, if space becomes an issue.
> >
> > The empirical results are interesting and provide a great future direction (as mentioned in my original review). Thus, I support the authors’ decision to include these results and arguments in the paper. The only concern here is the location where this argument is placed. Locating this argument right before the conclusion shifts the paper’s focus from the importance of tokenization to the utilization improvement of the experts, potentially blurring and weakening the main message of the paper. However, given the fact that the authors noted this suggestion and the suggestion is rather a minor presentation concern that does not affect the credibility of the main argument/results, it does not majorly affect the score negatively.
> >
> > > We thank the reviewer for providing formatting suggestions. …
> >
> > There are still multiple references that lack the publication year. Here is a summary of the errors:
> >
> > * Line 155, 158, 248, 359: Obando Ceron\* et al. → Obando Ceron\* et al. (2024)
> >
> > I highly recommend the authors amend these in-text citations in the camera-ready version.
> >
> > Overall, while there are still multiple minor technical/logical concerns, the authors' amendments in the revised version effectively addressed most of the major concerns raised in the review, resulting in a score increase.

---

### Meta-Review · Area_Chair_VGyy · 2024-12-21

**Metareview:**

This paper investigates what allowed soft-Mixture of experts (softMoEs) to scale performance with larger networks in a recent work. The authors argue with experimental analysis that tokenization of the encoder output in soft-MoEs played a significant role in their scalability, as demonstrated by tokenizing the encoder output and using a single expert. The work sheds light on what might be working well in recent work, although falls short of doing so comprehensively. Despite this the work still makes a step important for the community, and I recommend accepting it.

**Additional Comments On Reviewer Discussion:**

Four out of five reviewers highly appreciated the work by giving a score of 8, which was achieved in some cases by raising the score after thorough discussion during rebuttal. Reviewer XvpH mainly had concerns regarding the fairness of baseline scalability, which was sufficiently resolved during the discussion phase.

Reviewer XvpH’s concern regarding the role of specific tokenization in achieving results and de-emphasizing the role of MoEs is important. A discussion would be great for the future work on the possibility of “a more reasonable interpretation … that both tokenization and multiple experts contribute to SoftMoE’s performance, aligning with the fundamental concept of MoE rather than overemphasizing the importance of tokenization.”

---

### Decision · Program_Chairs · 2025-01-22

Accept (Spotlight)